**Subject Area:**
biochemistry

ZAG, AOC3, protein–protein interaction, cachexia, glycosylation

**Author for correspondence:**
Matthias Romauch
e-mail: matthias.romauch@gmail.com,
m.romauch@edu.uni-graz.at

# Zinc-α2-glycoprotein as an inhibitor of amine oxidase copper-containing 3

Matthias Romauch

Institute of Molecular Biosciences, Karl-Franzens-University, Graz, Austria

(iD) MR, 0000-0001-5586-1172

Zinc-α2-glycoprotein (ZAG) is a major plasma protein whose levels increase in chronic energy-demanding diseases and thus serves as an important clinical biomarker in the diagnosis and prognosis of the development of cachexia. Current knowledge suggests that ZAG mediates progressive weight loss through β-adrenergic signalling in adipocytes, resulting in the activation of lipolysis and fat mobilization. Here, through cross-linking experiments, amine oxidase copper-containing 3 (AOC3) is identified as a novel ZAG binding partner. AOC3—also known as vascular adhesion protein 1 (VAP-1) and semicarbazide sensitive amine oxidase (SSAO)—deaminates primary amines, thereby generating the corresponding alde-hyde, $H_2O_2$ and $NH_3$. It is an ectoenzyme largely expressed by adipocytes and induced in endothelial cells during inflammation. Extravasation of immune cells depends on amine oxidase activity and AOC3-derived $H_2O_2$ has an insulinogenic effect. The observations described here suggest that ZAG acts as an allosteric inhibitor of AOC3 and interferes with the associated pro-inflammatory and anti-lipolytic functions. Thus, inhibition of the deamination of lipolytic hormone octopamine by AOC3 represents a novel mechanism by which ZAG might stimulate lipolysis. Furthermore, experiments involving overexpression of recombinant ZAG reveal that its glycosylation is co-regulated by oxygen availability and that the pattern of glycosylation affects its inhibitory potential. The newly identified protein interaction between AOC3 and ZAG highlights a previously unknown functional relationship, which may be relevant to inflammation, energy metabolism and the development of cachexia.

## 1. Introduction

Zinc-α2-glycoprotein (ZAG) was first isolated from human plasma more than 50 years ago. Its name derives from its physico-chemical properties, as it precipitates with bivalent ions such as zinc, appears in the α2 fraction of electrophoretically separated plasma proteins and is glycosylated [1]. The highest expression levels of ZAG are found in liver [2,3], white adipose tissue [4,5] and prostate [6,7]. ZAG is primarily found in body fluids including plasma and semen, and is thought to mediate its effect by binding to the β3-adrenergic receptor [8]. ZAG is a major histocompatibility complex (MHC)-like molecule and accordingly its structure comprises a peptide-binding groove, surrounded by α-helices forming the α1 and α2 domains and the α3 subdomain [9,10]. Unlike classical MHC molecules, ZAG has no transmembrane domain and is therefore only found as a soluble protein in body fluids [11,12]. Furthermore, ZAG specifically binds fluorophore-tagged 11-(dansylamino)-undecanoic acid, which is not observed for other MHC homologues [11]. To date, only prolactin-inducible protein has been identified as a physiological ligand for seminal ZAG [12] but it is not clear whether ZAG forms part of the antigen-processing pathway.

ZAG has been associated with many divergent biological functions. For example, after stable transfection or addition to the medium, ZAG inhibits the progression of cancer cells through the cell cycle by downregulation of the

cyclin-dependent kinase 1 (CDK1) gene [13]. Intriguingly, the opposite effect was observed in 3T3-L1 pre-adipocytes: transfection with ZAG cDNA stimulated cell growth but inhibited differentiation, accompanied by a nearly 40% reduction in triglyceride content [14]. ZAG has also been identified as a ribonuclease, with comparable activity to onconase, but a much lower activity than RNase A [15]. In seminal fluid, ZAG is found on the surface of spermatozoa, where it is thought to be involved in sperm motility and capacitation [16,17].

ZAG is an important clinical marker in the diagnosis and prognosis of cancer [7,18]. It is strongly elevated in the plasma of cancer patients suffering from progressive weight loss [19,20]. Elevation of ZAG has been especially observed in patients suffering from cancer of the gastrointestinal system [21,22], breast [23,24] and prostate gland [7,18,25,26]. All these malignancies are accompanied by higher energy expenditure and progressive loss of muscle and fat mass [4,27,28]. This devastating state—named cachexia—is a multi-factorial syndrome that cannot be overcome by nutritional support and ultimately leads to functional impairment. The positive correlation between increased ZAG expression and weight loss has also been observed in mice suffering from tumour-induced cachexia [28–30]. ZAG is also elevated in chronic diseases of the heart [31], the kidney [32,33] and the lung [34–36], as well as in AIDS (acquired immunodeficiency syndrome) [37,38], all of which are also associated with the development of cachexia. However, ZAG levels are also significantly reduced during the early phase of sepsis, but increase again during recovery [39]. This is underpinned by the finding that ZAG is downregulated by pro-inflammatory mediators such as TNF-α: an inverse correlation between ZAG and TNF-α, VCAM-1, MCP-1 and CRP has been observed in patients suffering from systemic inflammation associated with chronic kidney disease, obesity and metabolic syndrome [40–42]. Therefore, ZAG is described as having an anti-inflammatory function.

ZAG has been also linked to the development of organ fibrosis [43]. An important mediator of this process is TGF-β, which turns fibroblasts into myofibroblasts, resulting in the production of large amounts of collagen and extracellular matrix components, thereby inducing dedifferentiation of surrounding parenchymal cells [44,45]. ZAG has been shown to counteract TGF-β-mediated effects [46]. Indeed, in experimental models of renal tubulointerstitial fibrosis, ZAG deficiency exacerbates deposition of interstitial collagen and fibroblast activation [43]. Furthermore, induction of cardiac hypertrophy and fibrosis in mice by thoracic aortic constriction leads to the same tissue alterations as interstitial fibrosis and fibroblast activation [43]. Notably, the exogenous application of recombinant ZAG reduces fibrosis in ZAG knockout (k.o.) mice to the level of heterozygous littermates. In vitro experiments revealed that TGF-β-induced expression of α-SMA can be blocked by addition of ZAG. Co-immunoprecipitation experiments showed that ZAG neither interacts with TGF-β nor its receptor, however. Furthermore, blocking ZAG signalling, which is supposedly mediated through the β3-adrenergic receptor, by propranolol, a non-selective antagonist of β-adrenergic receptors, did not restore TGF-β-induced α-SMA expression. This suggests that ZAG mediates its anti-inflammatory effect through a β3-adrenergic-independent signalling pathway [43].

ZAG-deficient mice exhibit mild obesity and reduced in vitro lipolysis. The lipolytic effect was tested by increasing cAMP levels using forskolin and isobutylmethylxanthine and stimulating β-adrenergic receptors using isoproterenol (β-nonspecific) and CL2316,243 (β3-specific). All tested substances showed reduced lipolysis compared with wild-type (wt) controls [47]. The authors suggest that ZAG might mediate its effect by binding to a receptor other than the β3-adrenergic receptor.

Taken together, ZAG seems to play many physiological roles, although scientists disagree on which signalling pathways mediate its effects. Hence, identifying the ZAG receptor could provide much-needed insight into the mechanism of ZAG function and stimulate future work in basic and clinical research on ZAG.

# 2. Results

## 2.1. ZAG binds to ectoenzyme AOC3

To attempt to identify ZAG interaction partners, purified recombinant ZAG and freshly prepared adipocyte plasma membranes were co-incubated and any physical interactions between them were stabilized by a photoactivatable cross-linker molecule (figure 1). Both human and murine ZAG (without leader sequence) were produced in E. coli after cloning in the expression plasmid pGEX-6P-2 and affinity purified by GST (glutathione-S-transferase)-tag. Both purified human and mouse proteins (GST-hZAG and GST-mZAG, respectively) and GST-tag alone—serving as a control—were labelled with the photoactivatable crosslinker Sulfo-SBED (Sulfo-N-hydroxysuccinimidyl-2-(6-[biotinamido]-2-(p-azido benzamido)-hexanoamido) ethyl-1,3′-dithioproprionate). Labelled GST-mZAG and GST-tag were incubated with prepared plasma membranes from murine wt adipose tissue, while GST-hZAG was incubated with plasma membrane from differentiated SGBS cells (human adipocyte cell line). After UV light exposure and the addition of β-mercaptoethanol (reducing agent), the samples were separated by SDS-PAGE and proteins revealed by western blot (WB) using streptavidin and anti-GST antibody. Using streptavidin, one band was detected using GST-tag (figure 2a(i), lane 1) as bait protein and three bands were detected using GST-mZAG or GST-hZAG as bait proteins (figure 2a(i), lane 2 and 3). The lowest band at approximately 26 kDa (kilodalton) represents the labelled GST-tag (*) (figure 2a(i), lanes 1–3) and was found in the control and samples incubated with GST-ZAG (**). This is due to loss of the GST-tag, which could not be completely prevented during overexpression of GST-ZAG in E. coli. The band at approximately 66 kDa represents labelled GST-ZAG (**) (figure 2a(i), lanes 2 and 3). The band at approximately 80 kDa represents a hitherto-unknown protein X (***), to which a biotin tag was transferred after reducing the cross-linker molecule with β-mercaptoethanol (figure 2a(i), lanes 2 and 3). Notably, the approximately 80 kDa band was only present when GST-mZAG or GST-hZAG were used as bait proteins. The GST-tag alone was not associated with any signal at approximately 80 kDa. Interestingly, using plasma membrane of SGBS cells (of human origin) led to the same signal as observed with murine wt adipocyte plasma membrane (figure 2a(i), lane 3). After stripping, the WB membrane was reprobed with α-GST antibody, when only GST-tag (figure 2a(ii), lane 1–3), GST-tagged murine ZAG (figure 2a(ii), lane 2) and GST-tagged human ZAG (figure 2a(ii), lane 3) were detected. Under non-reducing conditions (i.e. without

royalsocietypublishing.org/journal/rsob    Open Biol. 10: 190035

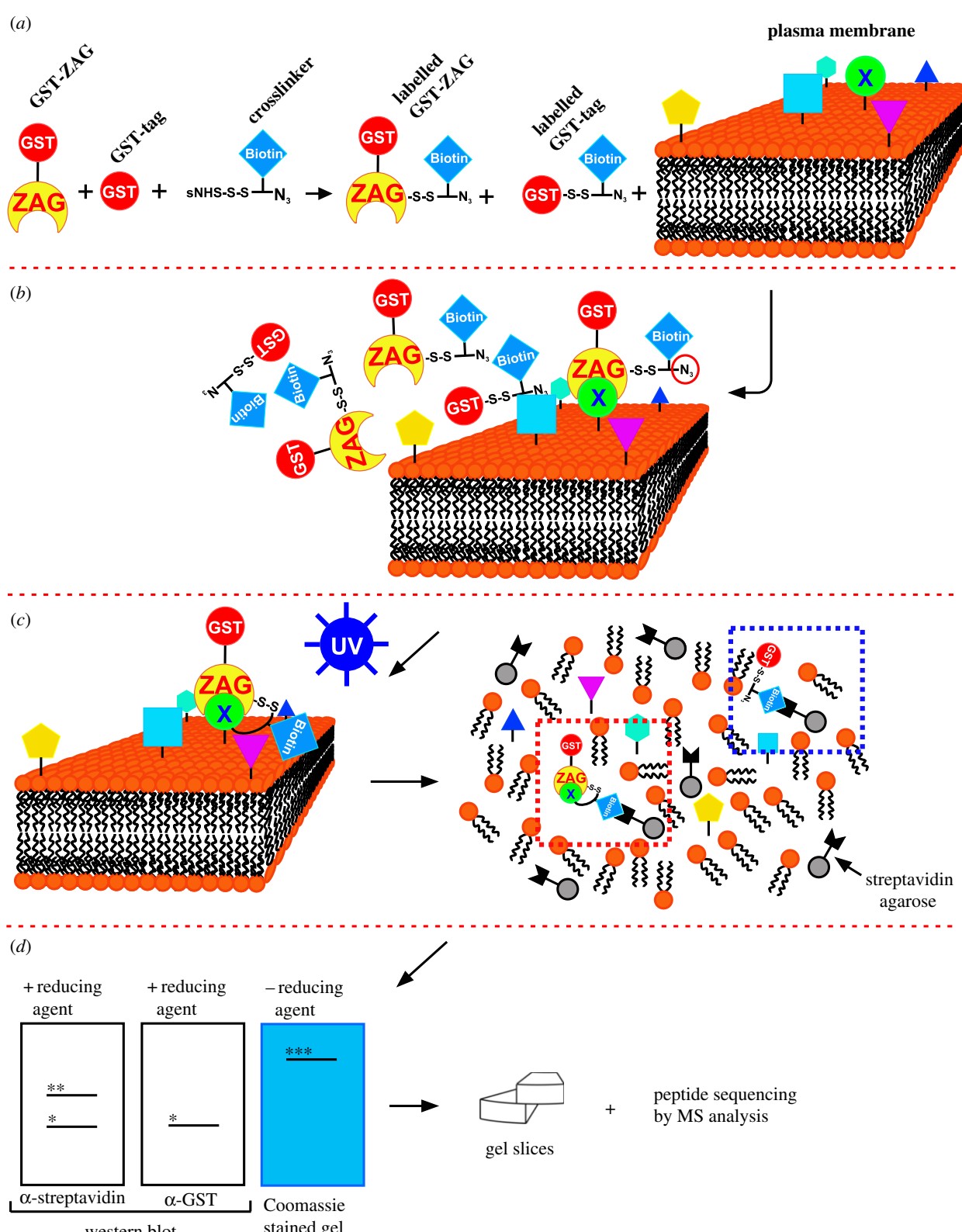

**Figure 1.** Synopsis of the cross-linking experiment. (*a*) First, murine and human GST-tagged ZAG and GST alone were overexpressed in *E. coli* and affinity purified. Plasma membranes were isolated from murine adipose tissue and SGBS cells. Purified proteins were labelled with Sulfo-SBED and co-incubated with isolated plasma membranes. (*b*) GST-ZAG binds to its interaction partner, whereas GST alone does not. (*c*) To stabilize the protein interaction, samples were exposed to UV light, inducing the highly reactive aryl azide (red circle, *b*) to form a covalent bond with a nearby amine. After cross-linking, the samples were delipidated and bound to streptavidin agarose via the biotin tag (blue square). The red-dotted box indicates the GST-ZAG/receptor complex, the blue-dotted box the GST-tag serving as a control. (*d*) Treated samples were separated by SDS-PAGE. Adding β-mercaptoethanol (reducing agent) split the disulfide bond leading to two bands, GST-ZAG (*) and the unknown protein (**). Without β-mercaptoethanol, a shift in MW of GST-ZAG was observed (***). For identification of the unknown interaction partner, samples were separated by non-reducing SDS-PAGE, stained with Coomassie Brilliant Blue and cut into pieces. Proteins extracted from gel slices were subjected to LC-MS/MS peptide sequencing.

β-mercaptoethanol and leaving the crosslinker uncleaved) the GST-ZAG signal (figure 2*a*(iii), lane 1) shifts to a size of around 250 kDa (figure 2*a*(iii), lane 2).

Due to the simplicity and availability of murine adipose tissue, special focus was placed on identifying the 80 kDa interaction partner. Non-reduced samples from the above

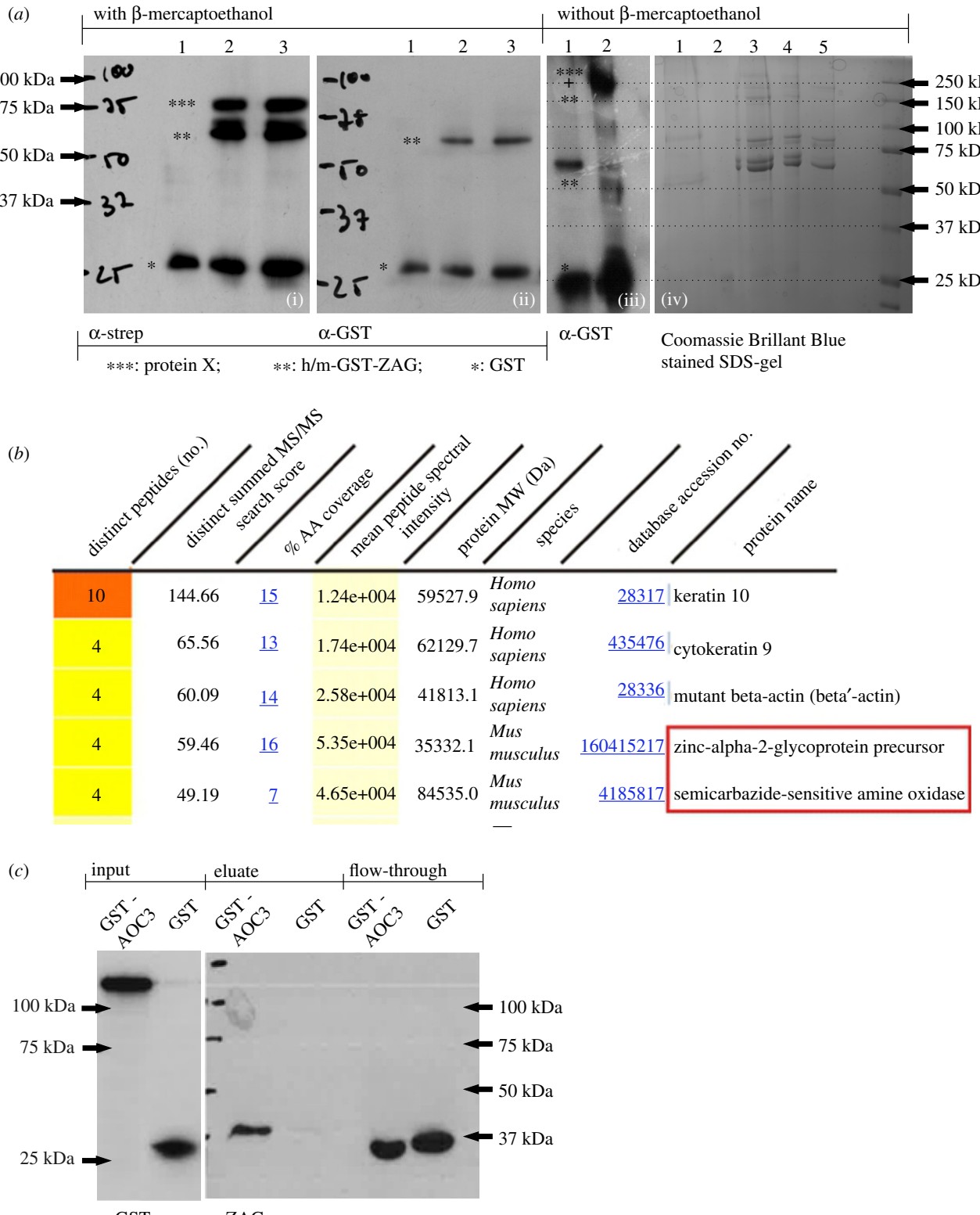

**Figure 2.** Analysis of cross-linking experiment. (*a*)(i) WB. Cross-linking samples carrying a biotin tag were bound to streptavidin agarose and eluted with 1 × SDS. Samples were reduced with β-mercaptoethanol and probed with streptavidin; lane 1: GST-tag incubated with plasma membrane of murine wt white adipose tissue; lane 2: GST-mZAG incubated with plasma membrane of murine wt white adipose tissue; lane 3: GST-hZAG incubated with plasma membrane of differentiated SGBS cells. (ii) WB. The membrane was stripped and probed with α-GST-antibody; lane 1: GST-tag incubated with plasma membrane of murine wt white adipose tissue; lane 2: GST-mZAG incubated with plasma membrane of murine wt white adipose tissue; lane 3: GST-hZAG incubated with plasma membrane of differentiated SGBS cells. (iii) WB. GST-mZAG (lane 1) and cross-linked GST-mZAG without β-mercaptoethanol (lane 2). (iv) Coomassie Brilliant Blue-stained SDS gel: proteins were separated by SDS-PAGE under non-reducing conditions. Corresponding bands were excised with a scalpel and prepared for peptide sequencing. Lane 1: plasma membrane; lane 2: plasma membrane with GST; lanes 3–5: plasma membrane with decreasing amounts of GST-mZAG. (*b*) Result of LC-MS/MS peptide sequencing: top five results of one band between 150 and 250 kDa (*a*(iv), lane 3). Besides keratin and actin, zinc-alpha2-glycoprotein and semicarbazide-sensitive amine oxidase sequences are found (red box). (*c*) WB of GST-pulldown: GST and murine GST-AOC3 were purified from lentivirally transduced HEK293 cells and incubated with plasma of C57Bl6 wt mice. After performing the GST-pulldown experiment proteins were separated by SDS-PAGE and blotted proteins detected using α-GST and α-ZAG antibody.

royalsocietypublishing.org/journal/rsob Open Biol. **10**: 190035

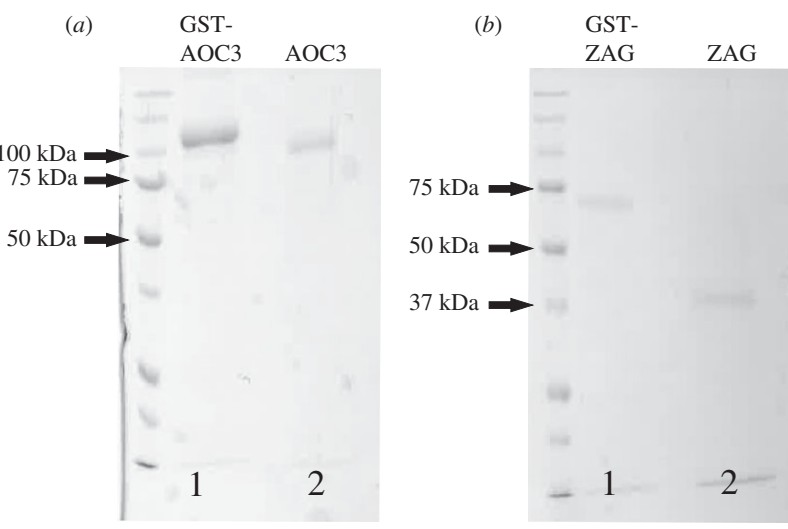

**Figure 3.** Coomassie Brilliant Blue-stained SDS gel: GST-tagged AOC3 (*a*, lane 1) and GST-tagged ZAG (*b*, lane 1) were affinity purified from the conditioned medium of lentivirally transduced HEK293 cells. The GST-tag was removed by PreScission Protease (*a* and *b*, lane 2).

affinity purification were separated by native SDS-PAGE to guarantee their presence in the same gel fraction. The gel was stained with Coomassie Brilliant Blue and bands excised with a scalpel (figure 2*a*(iv)). For orientation, a WB of non-reduced samples probed with streptavidin was carried out in parallel. Excised bands were prepared and subjected to mass spectrometry-based peptide sequencing. One excised band contained ZAG and identified SSAO (figure 2*b* red box)—from this point named AOC3—as a putative interaction partner. AOC3 has a molecular weight of approximately 84 kDa and exists as a homodimer. Given this, the shift of the GST signal to a higher molecular weight under non-reducing conditions (figure 2*a*(iii)) can be explained by binding between one homodimeric AOC3 and at least one GST-ZAG molecule. To confirm the newly identified protein interaction, it was attempted to purify AOC3 from *E. coli*. Since all expression conditions failed, a modified method using HEK293 cells as expression host was chosen [48]. Using lentivirus, secretable forms of GST-AOC3 (figure 3) and GST-tag were stably expressed in HEK293 cells. Both proteins were affinity purified from the conditioned medium. To ascertain whether recombinant GST-AOC3 interacts with murine plasma ZAG, a GST-pulldown was performed (figure 2*c*). Plasma of overnight-fasted C57Bl6 male wt mice was incubated with recombinant GST-AOC3 and GST as a control. A WB of the eluate fraction revealed that GST-AOC3 bound ZAG from murine plasma, whereas GST alone did not.

## 2.2. ZAG functions as an allosteric inhibitor of AOC3

AOC3 belongs to the family of copper-containing amine oxidases. It catalyses the oxidative deamination of primary amines, generating the corresponding aldehydes, hydrogen peroxide ($H_2O_2$), and ammonia ($NH_3$). The enzyme forms a homodimer, with each unit bound to the plasma membrane via a short transmembrane domain and the catalytic centre oriented on the extracellular side [49]. For activity measurements, recombinant or endogenous AOC3 is incubated with the synthetic substrate benzylamine or [$^{14}$C]-benzylamine. Non-radioactive assays measure $H_2O_2$, which oxidizes Amplex Red to its fluorescent analogue resorufin in the presence of horseradish peroxidase (HRPO). Using

[$^{14}$C]-benzylamine as substrate, the activity corresponds to the amount of [$^{14}$C]-benzaldehyde generated. For each molecule of benzylamine, one molecule of $H_2O_2$ and one molecule of $NH_3$ are generated. LJP1586 (Z-3- fluoro-2-(4-methoxybenzyl)-allylamine hydrochloride) serves as an inhibitor. To investigate whether ZAG can modulate AOC3 activity, both GST-tagged AOC3 and ZAG of murine origin were purified from lentivirally transduced HEK293 cells and the GST-tag was removed. In all control assays, ZAG was replaced by the same amount of GST purified from stably transfected HEK293 cells. In a first attempt, activity assays were performed using Amplex Red reagent (figure 4*a*). A saturation curve using benzylamine as substrate revealed the highest activity at 100 μM (figure 4*b*). To characterize the interaction between AOC3 and ZAG, both proteins were mixed at different molar ratios. A stepwise increase in the concentration of recombinant ZAG led to a stepwise decrease in recombinant AOC3 activity. The strongest inhibition was observed at a molar AOC3/ZAG ratio of approximately 1 : 1 (25 ng ZAG) (figure 4*c*). GST alone did not show any inhibitory effect. Next, the mechanism of inhibition was investigated by generating a Michaelis–Menten plot, which revealed that $V_{max}$ (maximum velocity) decreases, whereas $K_m$ (i.e. the Michaelis–Menten constant: substrate concentration at half-maximum velocity) remains almost constant, with rising ZAG concentrations (figure 4*d*). A Lineweaver–Burk diagram clearly illustrates the difference in $V_{max}$ and $K_m$ behaviour. A constant $K_m$ value is represented by the intersection of the function with the *x*-axis (figure 4*e*). This suggests that ZAG functions as a highly effective allosteric inhibitor of AOC3. It means that ZAG binds AOC3, but not at the catalytic site, thereby reducing the activity of the enzyme in a non-competitive manner.

Subsequently, it was tested whether recombinant mammalian ZAG inhibits endogenous AOC3 activity as effectively as that of recombinant AOC3. Since AOC3 is expressed on the surface of adipocytes and endothelial cells, differentiated 3T3-L1 adipocytes and human coronary artery endothelial cells (HCAEC) were chosen. As a positive control for inhibition, AOC3 activity was blocked by inhibitor LJP1586. To eliminate any non-specific background signals in the cell experiments, assays were performed radioactively. The activity of 3T3-L1-derived AOC3 was effectively reduced as

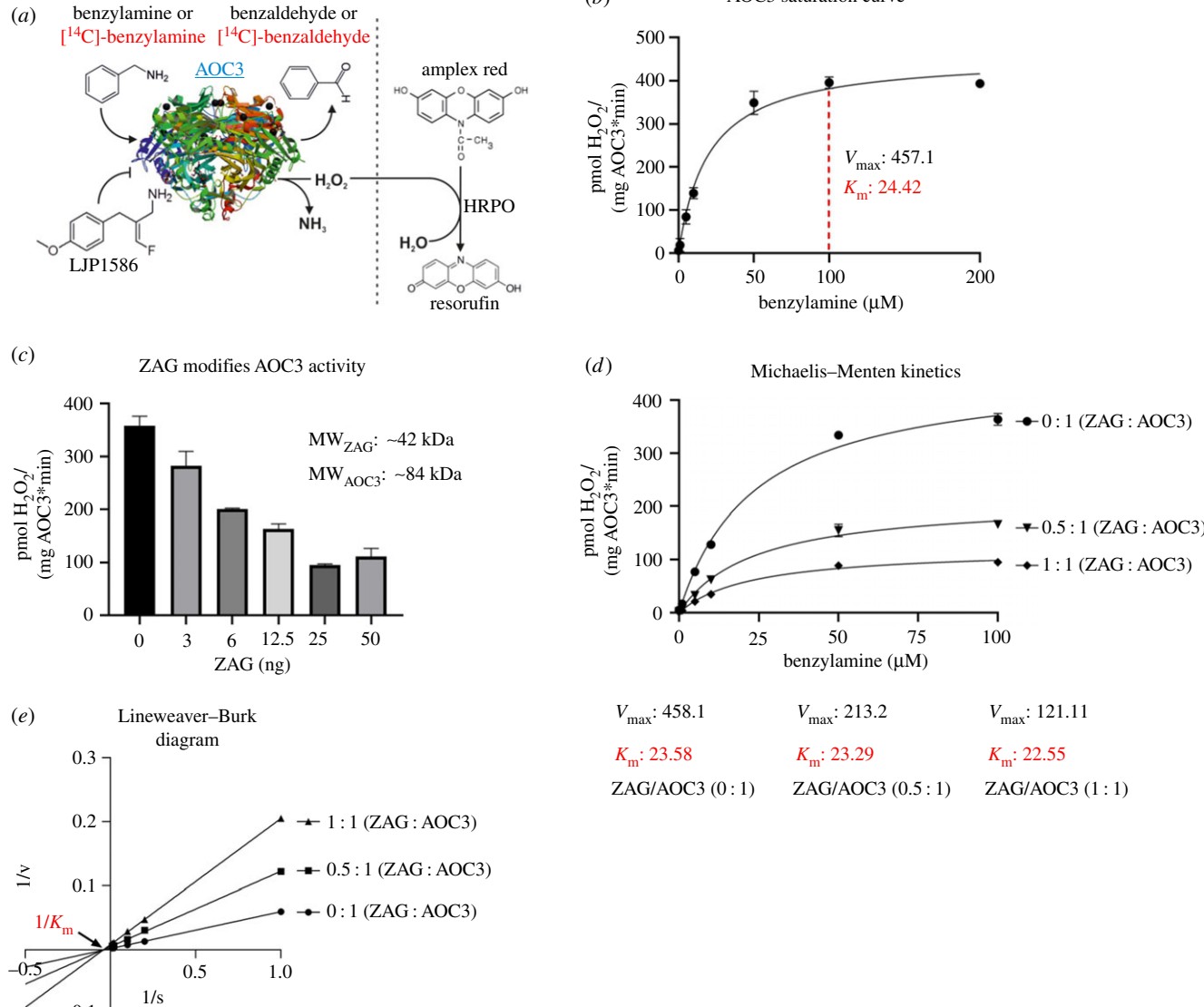

**Figure 4.** Enzyme kinetics. (*a*) Illustration of AOC3 activity measurement. LJP1586: inhibitor; HRPO: horse radish peroxidase; crystal structure of AOC3 modified from RCSB PDB, PDB-ID: 2C10 [49]. (*b*) AOC3 saturation curve: activity of AOC3 (50 ng) at different concentrations of benzylamine. Highest activity indicated by red dashed line (100 μM). $V_{max}$: maximum velocity; $K_m$: Michaelis–Menten constant. (*c*) AOC3/ZAG activity assay. AOC3 (50 ng) and ZAG were mixed at different molar ratios. Molecular weights are 42 kDa for ZAG ($MW_{ZAG}$) and 84 kDa for AOC3 ($MW_{AOC3}$). Assays and control contained the same amount of AOC3. (*d*) Michaelis–Menten plot. AOC3 (50 ng) and ZAG, mixed at different ratios, were incubated at different substrate concentrations. $V_{max}$: maximum velocity; $K_m$ = Michaelis–Menten constant. (*e*) Lineweaver–Burk diagram: allosteric inhibition illustrated by intersection of functions with x-axis at the same point (constant $K_m$: Michaelis–Menten constant).

the amount of recombinant ZAG was increased. The addition of 50 μg ml$^{-1}$ recombinant ZAG inhibited [$^{14}$C]-benzaldehyde formation to a similar extent as LJP1586 (figure 5*a*). This is remarkable since the highest concentration of ZAG used (50 μg ml$^{-1}$ = approx. 1.2 μM) is nearly tenfold less in molar terms than for the small molecule inhibitor LJP1586 (10 μM). This underpins the highly specific nature of this protein inter-action, with similar ZAG concentrations being present in human plasma (approx. 50 μg ml$^{-1}$ serum) [50]. Furthermore, the inhibitory effect of recombinant ZAG on HCAEC AOC3 (figure 5*b*) confirmed the similarity between murine and human AOC3, underlining the cross-linking results obtained with SGBS cell membranes and indicating that the ZAG–AOC3 interaction also plays an important role in humans. Since the inhibitor LJP1586 is designed for murine AOC3, a higher concentration was needed to block human AOC3 of HCAEC origin. Comparing the raw data, 3T3-L1 adipocytes and HCAEC cells showed the same AOC3 activity. The ten-fold-higher activity of 3T3-L1 adipocytes compared with

HCAEC relates to the normalization to mg cellular protein/ measurement: 3T3-L1 adipocytes contain much less protein.

Since recombinant ZAG inhibits endogenous AOC3, it was asked whether endogenous ZAG could inhibit recombi-nant AOC3. For this purpose, plasma of C57Bl6 wt mice was collected and rebuffered in 10 mM Tris-HCl, pH 8 (ZAG-pI: approx. 5.8). Plasma proteins were separated by anion exchange chromatography and eluted by linear NaCl gradi-ent (figure 6*a*). ZAG-containing fractions were identified by WB and used for activity assays. Fractions C12 and D1 showed a signal between 37 and 50 kDa, which corresponds to murine plasma ZAG (figure 6*a*). ZAG-containing fractions (C12 and D1) were pooled, as were fractions without any ZAG (D3 and D4) as controls, and incubated with 50 ng recombinant AOC3 (figure 6*b*). The ZAG-positive fractions reduced recombinant AOC3 in a dose-dependent manner. However, the control IEX fractions, which contained no ZAG, enhanced recombinant AOC3 activity in a dose-dependent manner, rather than having the expected neutral

royalsocietypublishing.org/journal/rsob   Open Biol. **10**: 190035

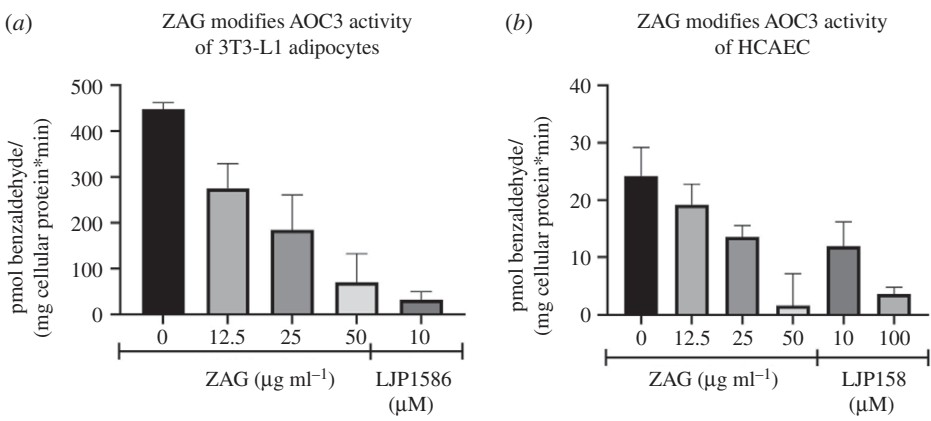

**Figure 5.** (*a,b*) [$^{14}$C]-benzylamine assay: differentiated 3T3-L1 cells (*a*) and HCAECs (*b*) were incubated with increasing amounts of recombinant ZAG. In parallel, cell-derived AOC3 activity was blocked by adding LJP1586.

**Figure 6.** AOC3-inhibitory effect of plasma-derived ZAG. (*a*) IEX elution diagram and WB. Murine plasma of C57Bl6 mice was collected and rebuffered in 10 mM Tris HCl, pH 8. Plasma was separated by ion exchange chromatography (IEX) and eluted by linear NaCl gradient. ZAG-containing fractions were identified by WB using α-ZAG antibody. (*b*) [$^{14}$C]-benzylamine assay. ZAG-IEX fractions and no ZAG-IEX fractions were incubated with recombinant AOC3 (50 ng). (*c*) IEX elution diagram and WB. Comparison of IEX diagram and WB of wt and ZAG k.o. plasma. ZAG-containing fractions were identified by WB using α-ZAG antibody. (*d*) [$^{14}$C]-benzylamine assay. IEX fractions (C12 and D1) of wt mice and corresponding fractions of ZAG k.o. mice were incubated with recombinant AOC3 (50 ng). Data are presented as mean ± s.d.: ***$p < 0.001$.

effect. This stimulatory effect is probably due to both endogenous AOC3 activity and plasma components. First, murine plasma (except that of AOC3 k.o. mice) contains endogenous amine oxidase activity, which can be blocked by the inhibitor LJP1586 (electronic supplementary material, figure S1A–C). Plasma-derived AOC3 activity results from

release of the membrane-bound enzyme from cells by metalloprotease activity [51–53]. However, measurement of amine oxidase activity of IEX fractions—either containing or not containing ZAG—before adding recombinant AOC3 revealed no endogenous activity (electronic supplementary material, figure S2C). Second, incubation of recombinant AOC3 with

royalsocietypublishing.org/journal/rsob  Open Biol. **10**: 190035

plasma of wt, AOC3 k.o. and ZAG k.o. mice enhanced AOC3 activity approximately threefold (electronic supplementary material, figure S1D). Therefore, a plasma component found in all three genotypes must be responsible for enhancing AOC3 activity. Third, the IEX fractions lacking ZAG (D3 and D4) correspond to the major protein peak of the IEX profile, which mostly derives from albumin. Incubating recombinant AOC3 with fatty acid-free bovine serum albumin (BSA) also enhanced recombinant AOC3 activity to the same extent as murine plasma (electronic supplementary material, figure S2A). Fourth, the literature describes a low-molecular-weight plasma component (3.8 kDa), which in combination with lysophosphatidylcholine (LPC) boosts AOC3 activity [54]. LPC makes up to 4–20% of total plasma phospholipid content [55] and albumin is an important LPC storage protein [56]. Therefore, the IEX fractions lacking ZAG may contain the AOC3-activating plasma component, which is fully active in the presence of LPC. Notably, incubation of human lung-microsomal AOC3 with filtered and lyophilized human plasma (FLHP) enhances AOC3 activity up to fivefold [54], which is similar to the effect of adding 200 µl of the IEX fractions lacking ZAG to recombinant AOC3 (figure 6b).

To substantiate this finding, the plasma of wt mice and ZAG k.o. mice were compared. ZAG-containing fractions (C12 and D1) of wt plasma were identified by WB using α-ZAG antibody (figure 6c). A WB of the corresponding fractions of ZAG k.o. plasma showed no signal (figure 6c). Fifty µl of the IEX fractions of wt plasma and the corresponding IEX fractions of ZAG k.o. plasma were incubated with 50 ng AOC3. As before, ZAG-containing fractions of wt plasma reduced benzylamine catalysis by AOC3, whereas the corresponding fractions of ZAG k.o. plasma did not (figure 6d).

## 2.3. ZAG inhibition of AOC3 augments stimulation of lipolysis

Most of the literature describes ZAG as an agonist of the β-adrenergic receptors, thereby stimulating downstream elements leading to an increase in lipolysis. To test this hypothesis, ZAG (50 µg ml$^{-1}$), GST (50 µg ml$^{-1}$), LJP1586 (10 µM) and isoproterenol (10 µM), a short-acting non-specific β-adrenergic agonist, were incubated with differentiated 3T3-L1 cells (figure 7a). Compared to ZAG, isoproterenol significantly enhanced glycerol release already within the first thirteen minutes. ZAG, GST and LJP1586 showed no such effect. Incubating differentiated 3T3-L1 cells with ZAG (50 µg ml$^{-1}$), GST (50 µg ml$^{-1}$) and LJP1586 (10 µM) for several hours revealed that, although ZAG showed a lipolytic effect, this did not occur until 12 h (figure 7b). Therefore, ZAG definitely does not behave as a classical β-adrenergic agonist such as isoproterenol and another mechanism must be involved in ZAG-stimulated lipolysis, most likely involving AOC3. Although it is not well characterized, AOC3 is thought to be involved in the catalysis of biogenic amines such as methylamine, aminoacetone, dopamine, histamine and trace amines [57,58]. Hence, blockade of AOC3-dependent deamination of biologically active amines by ZAG might indirectly affect the metabolic state of the cell. To investigate which biogenic amines might modulate lipolytic activity, a set of biogenic amines was tested for catalysis by AOC3 (figure 7c). A colorimetric assay based on 4-nitrophenyl-boronic acid

oxidation in the presence of $H_2O_2$ was performed [59], since molecules such as noradrenaline, octopamine and dopamine interfered with the Amplex Red assay, due to the photo-oxidation of substrates [60]. The strongest activity was observed with tyramine, histamine, dopamine, cadaverine, cysteamine, ethanolamine, octopamine, putrescin, spermidine, isopentyla-mine and benzylamine (figure 7c). Subsequently, the same set of biogenic amines was tested for their ability to stimulate lipolysis in 3T3-L1 adipocytes (figure 7d). Comparing the two assays revealed that histamine, cysteamine, cadaverine and octopamine (trace amine) are converted by AOC3 and stimulate lipolysis to a varying degree. Notably, noradrenaline and octopamine both strongly stimulated lipolysis, but only octopamine was converted by AOC3. In the follow-up experiments, only noradrenaline and octopamine were used to generate a significant difference in glycerol release. Noradrenaline belongs to the family of catecholamines and is described as an agonist of α- and β- adrenergic receptors [61]. Octopamine belongs to the family of trace amines and functions by binding to trace amine associated receptor 1 (TAAR1) and β3-adrenergic receptor [62,63]. To ask whether reduced AOC3 activity enhances glycerol release, lipolysis stimulation assays were performed in the presence of LJP1586 or ZAG. In all control assays, ZAG was replaced by the same amount of GST purified from stably expressing HEK293 cells. In the case of noradrenaline and isoproterenol, the addition of LJP1586 did not enhance glycerol release (figure 7e,f). This is in line with the observations that noradrenaline is not converted by AOC3 and isoproterenol contains no primary amine. However, the presence of LJP1586 (10 µM) or ZAG (50 µg ml$^{-1}$) boosted octopamine-stimulated lipolysis (figure 8a,b). This suggests that reduced deamination of octopamine results in higher bioavailability, leading to a stronger β3-adrenergic or possibly TAAR1-mediated lipolytic stimulation, although the presence of TAAR1 in adipocytes has not been described thus far. Finally, it is notable that the effect of ZAG diminished with increasing octopamine concentration compared with LJP1586 (figure 8a,b), which probably points to a different mode of inhibition.

## 2.4. The inhibitory potential of recombinant ZAG depends on glycosylation

ZAG is a highly abundant protein found in body fluids such as blood and semen. According to the literature, ZAG can be glycosylated in a number of different ways, suggesting different functions [64,65]. This is in accordance with the observation that the plasma ZAG of different C57Bl6 mice was not always the same size (figure 9a1). The notion that this size difference depends on glycosylation was proven by the treatment of murine plasma proteins with PNGase F (peptide: N-glycosidase F). Upon treatment with PNGase F, ZAG reduced in size to approximately 32 kDa (calculated MW approx. 33.6 kDa) according to SDS-PAGE. In ZAG k.o. plasma, no ZAG signal was detected (figure 9a2). During this study, recombinant ZAG expression was tested in different expression hosts such as *E. coli*, *Saccharomyces cerevisiae*, *Komagatella pastoris*, *Sf9* and *BTI-Tn-5B1-4* insect cells, as well as the mammalian cell lines Expi293F and adherent HEK293 cells. Among all the expression hosts tested, the largest difference in glycosylation was found between Expi293F and HEK293 cells. Expi293F cells are HEK293 cells adapted to grow in suspension; they are used for large-scale production

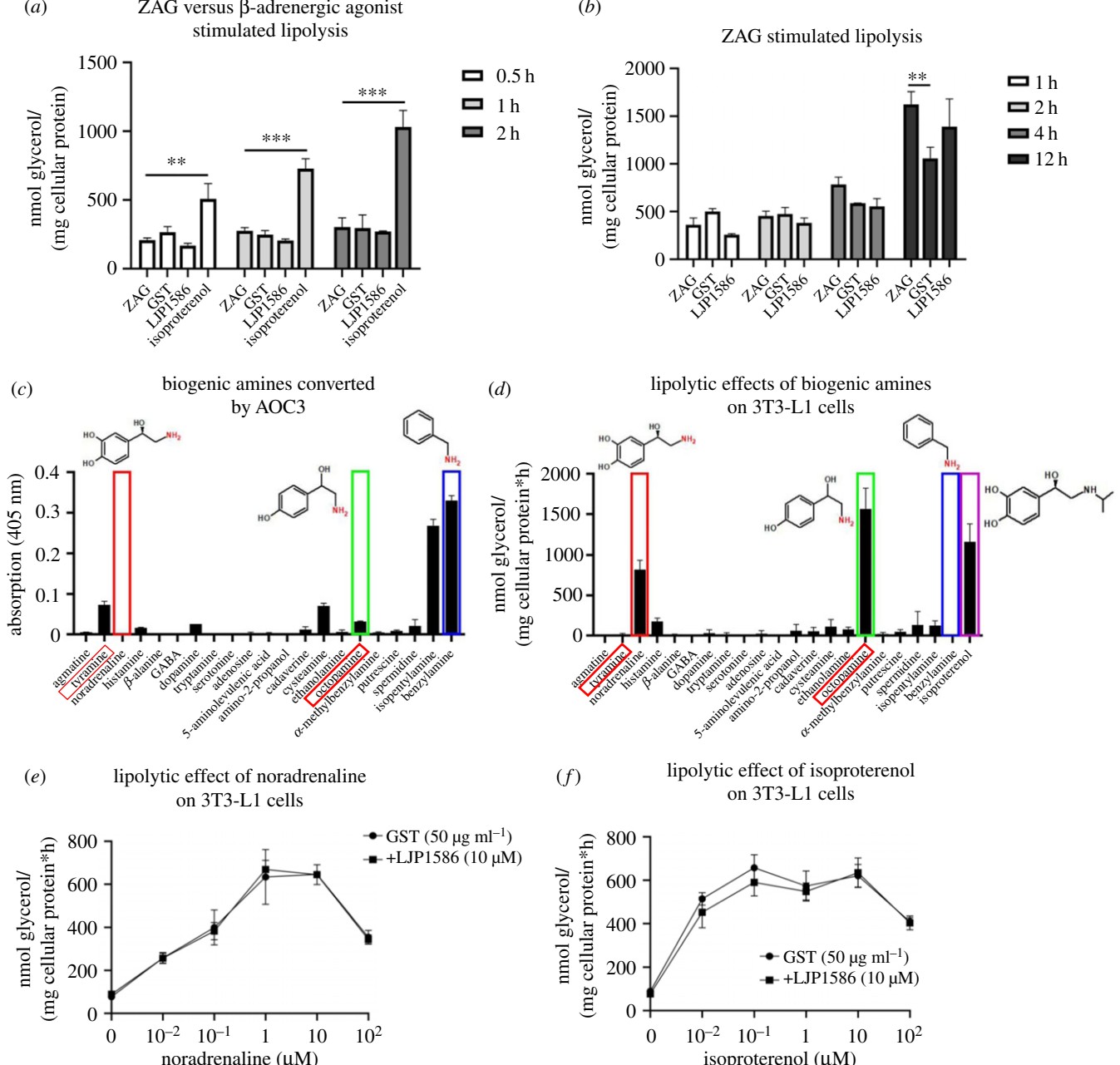

**Figure 7.** (a) Comparison of ZAG- and β-adrenergic agonist-stimulated lipolysis. Fully differentiated 3T3-L1 cells were incubated with ZAG (50 µg ml⁻¹), GST (50 µg ml⁻¹), LJP1586 (10 µM) and isoproterenol (10 µM). Glycerol release was monitored for 2 h. (b) ZAG-stimulated lipolysis. Fully differentiated 3T3-L1 cells were incubated with ZAG (50 µg ml⁻¹), GST (50 µg ml⁻¹) and LJP1586 (10 µM). Glycerol release was monitored for 12 h. (c) Screen for biogenic amines converted by AOC3. A set of biogenic amines (20 mM) was tested for deamination by recombinant AOC3. Activity was measured by 4-nitrophenyl-boronic acid oxidation. Red boxes around names indicate trace amines. (d) Screen for biogenic amines stimulating lipolysis. Fully differentiated 3T3-L1 adipocytes were incubated with the same set of biogenic amines. Lipolytic activity was measured by glycerol release. Red boxes around names indicate trace amines. (e,f) Glycerol release from 3T3-L1 cells. Lipolytic activity of noradrenaline and isoproterenol in the presence and absence of LJP1586 was tested. Data are presented as mean ± s.d.: **$p < 0.01$; ***$p < 0.001$.

of recombinant proteins in industry. Overexpressing GST-ZAG in HEK293 cells and removal of the GST-tag by Pre-Scission Protease resulted in a SDS-PAGE band of less than 46 kDa (figure 9a3). Overexpressing GST-ZAG and Flag-ZAG in Expi293F resulted in a broad band that became more diffuse with increasing expression time (figure 9a4). To confirm that this was due to glycosylation, Expi293F-driven GST-ZAG overexpression was combined with tunicamycin (a compound suppressing glycosylation in general) at different concentrations (figure 9a5). Overexpression of GST-ZAG and Flag-ZAG in the presence of 1 µg ml⁻¹ tunicamycin showed a reduction in size of both proteins (figure 9a6). Treating Expi293F-derived GST-ZAG with PNGase F led to

the same result. After removal of the GST-tag, two distinct bands (figure 9a7, asterisks) were detected and, after deglycosylation by PNGase F, both ZAG bands appeared to combine at approximately 32 kDa (figure 9a7), as observed with PNGase F-treated murine plasma ZAG and tunicamycin-treated Flag-ZAG. Since O-glycosylation might also affect the size of the protein, another overexpression experiment was performed with GST-ZAG in the presence of the inhibitor benzyl-2-acetamido-2-deoxy-α-D-galactopyranoside (BAGN). No effect on the size of the protein was observed, however, which underlines the notion that size differences depend on N-glycosylation events (figure 9b). The N-glycosylation site—known as the sequon—is defined by the amino acid

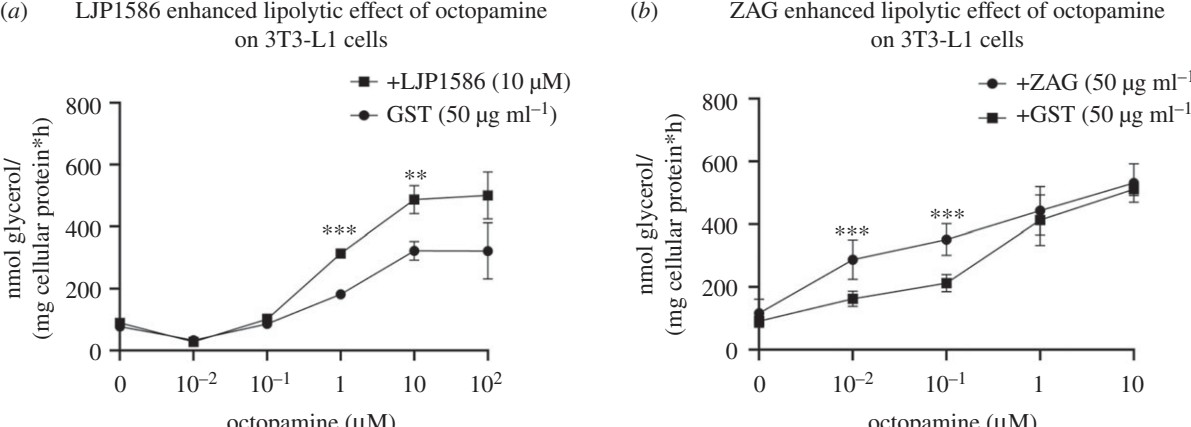

**Figure 8.** Stimulated glycerol release from 3T3-L1 cells in the presence of LJP1586 and ZAG. (a,b) Direct comparison of octopamine-stimulated lipolysis in the presence of LJP1586 (10 µM) and ZAG (50 µg ml⁻¹). Corresponding amounts of GST (26 kDa) purified from HEK293 cells served as control. Data are presented as mean ± s.d.: **$p < 0.01$; ***$p < 0.001$.

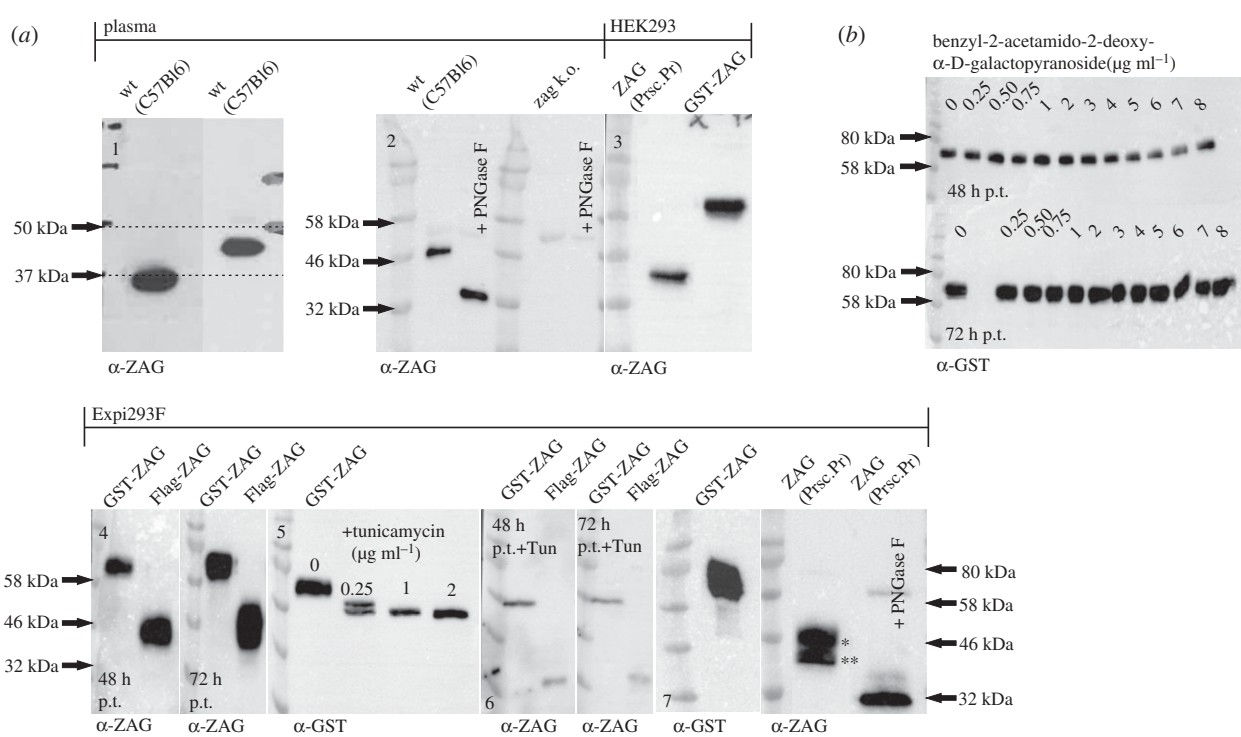

**Figure 9.** Glycosylation of ZAG. (a) WB. (1) Differences in size of ZAG in individual C57Bl6 wt mice. (2) Effect of PNGase F treatment on plasma ZAG of wt and ZAG k.o. C57Bl6 mice. (3) Murine ZAG overexpressed in HEK293 cells with and without GST-tag. (4) Overexpression of GST-ZAG and Flag-ZAG in Expi293F cells. Samples were collected after 48 and 72 h post transfection (p.t.). (5) Overexpression of GST-ZAG in Expi293F cells in the presence of different concentrations of tunicamycin. (6) Overexpression of GST-ZAG and Flag-ZAG in Expi293F cells in the presence of tunicamycin (1 µg ml⁻¹). Samples were collected after 48 and 72 h p.t. (7) Overexpression of GST-ZAG in Expi293F cells and sequential treatment with PreScission Protease and PNGase F. (*) and (**) indicate different glycoforms. (b) WB. Overexpression of GST-ZAG in presence of the O-glycosylation inhibitor benzyl-2-acetamido-2-deoxy-α-D-galactopyranoside. Samples were collected after 48 and 72 h p.t. Proteins were detected using α-ZAG or α-GST antibody.

sequence Asn-X-Ser (asparagine-X-serine) or Asn-X-Thr (asparagine-X-threonine). X can be any amino acid except proline and the Asn residue serves as the anchor point for N-glycosylation. The murine ZAG peptide sequence has three different N-glycosylation sites at positions 123, 190 and 254 (the numbers relate to the position of the Asn residue within the murine peptide sequence with the leader sequence). Different glycoforms of ZAG were generated by mutating the Asn residues to glutamine. Single mutations and combined mutations led to seven different ZAG glycoforms: 123, 190, 254, 123/190, 123/254, 190/254 and Δ3 (where, in the latter

case, all three sites were mutated). Using HEK293 and Expi293F cells as expression hosts showed that the size of the protein declined with the number of available N-glycosylation sites (figure 10a1 and b1). Nevertheless, reducing the number of glycosylation sites did not lead to a discrete, monodisperse ZAG band in Expi293F cells, as was observed for ZAG overexpressed in HEK293 or for plasma-derived ZAG. One sample of Expi293F cells collected 24 h post-transfection gave a more disperse signal (figure 10a2). The molecular weights of Expi293F-derived ZAG result from differently glycosylated isoforms (asterisks). It appears that overexpression in Expi293F

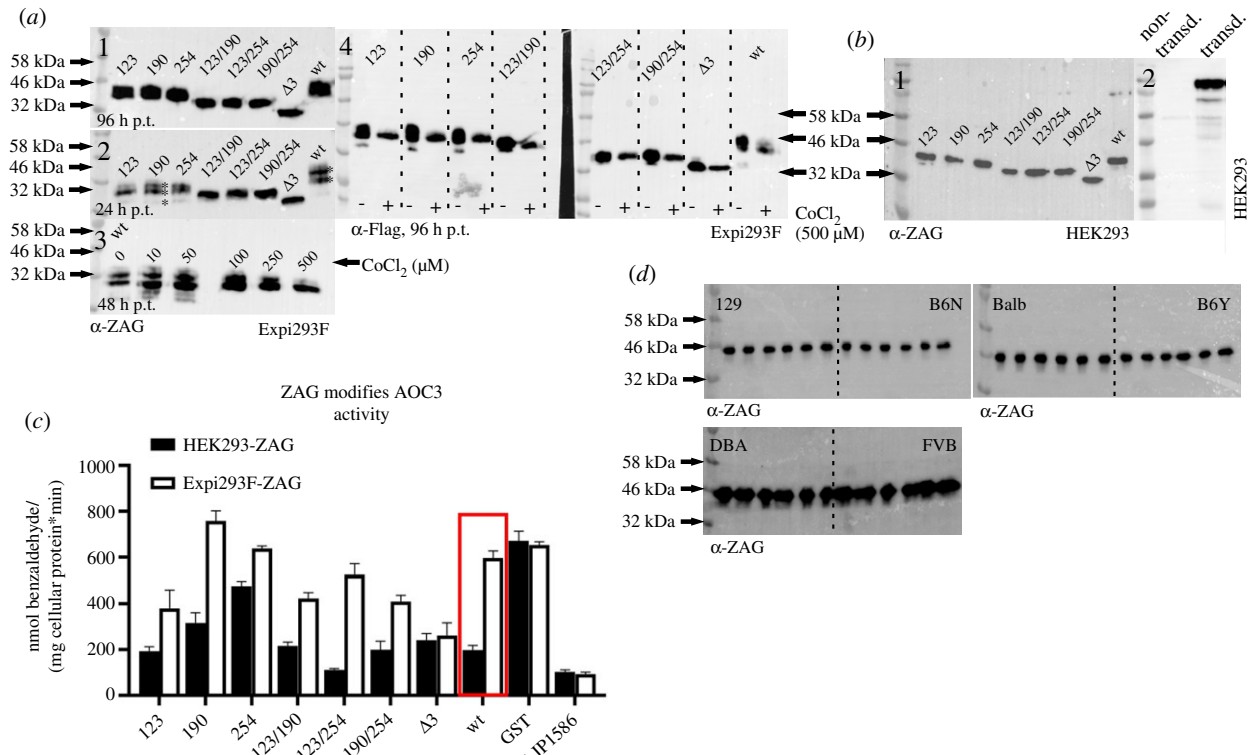

**Figure 10.** Glycosylation of ZAG. (a) WB. (1) Overexpression of wt and glycomutants of Flag-ZAG in Expi293F cells. Samples were collected after 96 h post transfection (p.t.). (2) Overexpression of wt and glycomutants of Flag-ZAG in Expi293F cells. Samples were collected after 24 h p.t. Asterisks (*) indicate different glycoforms. (3) Overexpression of wt Flag-ZAG in Expi293F cells in the presence of different concentrations of the hypoxia mimetic CoCl₂. Samples were collected after 48 h p.t. (4) Expression of wt and glycomutants of Flag-ZAG in the presence of 500 µM CoCl₂. Samples were collected after 96 h p.t. (b) WB. (1) Wt and glycomutants of ZAG overexpressed in HEK293 cells. (2) HEK293 cells lentivirally transduced (transd.) with full-length AOC3 (i.e. including transmembrane domain). (c) [¹⁴C]-benzylamine assay. Inhibitory potential of wt and glycomutants of ZAG overexpressed in HEK293 and Expi293F cells. Wt and ZAG glycomutants were purified from HEK293 and Expi293F cells and incubated with HEK293 cells stably expressing murine AOC3 (b, 2). (d) WB. Plasma proteins (5 µg) of mouse strains 129, B6N, Balbc, B6Y, DBA and FVB were separated by SDS-PAGE. For each mouse strain, plasma was taken from six different male mice (greater than 12 months old). Proteins were detected using α-Flag or α-ZAG antibody.

cells leads to one higher MW and one lower MW glycosylated form in addition to the 'true' isoform.

This variability in the same cell line might be due to growth conditions, which can affect post-translational modifications. Thus, HEK293 cells are adherent and grow in serum containing medium, whereas Expi293F cells grow in serum-free suspension. Concerning serum as medium supplement, serum-free medium is already described to significantly increase N-linked glycosylation of interleukin-2 when overexpressed in suspension growing baby hamster kidney cells [66]. Adding serum at different concentrations to Expi293F suspension culture made the cells clump and did not change glycosylation. In other types of suspension culture with host cells such as *E. coli*, insect cells (*BTI-Tn-5B1-4* and *Sf9*) or mammalian cells, oxygenation has a major impact on the success of protein expression [67–69]. From this perspective, it seemed likely that the state of oxygenation might influence the glycosylation pattern of ZAG. Since Expi293F cells grow in suspension, they might have a higher level of oxygenation than HEK293 cells. Hence, reducing oxygenation might simplify the ZAG glycosylation pattern. To test this hypothesis, Expi293F cultures were supplemented with CoCl₂, a hypoxia mimetic substance. Hypoxia is transcriptionally co-mediated by the transcription factor HIF1-α. During normoxia, HIF1-α is prolyl hydroxylated by prolyl-4-hydroxylases (PHDs), directing the protein to degradation by ubiquitylation. Hypoxia induces the opposite: HIF1-α is stabilized and PHDs are inhibited [70,71].

CoCl₂ mimics hypoxia by inhibiting HIF1-α hydroxylation by PHDs. In a first attempt, different concentrations of CoCl₂ were tested. Indeed, supplementing the media with the highest concentration of CoCl₂ (500 µM) simplified the signal of wt ZAG overexpressed in Expi293F cells from multiple bands to a single band, which is similar to the appearance of wt ZAG when expressed in HEK293 cells (figure 10a3). In another experiment, all glycomutants of ZAG were overexpressed in Expi293F cells in the presence of 500 µM CoCl₂. As observed for the wt form, all ZAG mutants appeared as a single band (figure 10a4).

To investigate how glycosylation affects the inhibition of AOC3 by ZAG, all glycomutants were overexpressed in HEK293 and Expi293F cells in parallel. After GST affinity purification, PreScission Protease digestion and dialysis, all proteins were adjusted to a concentration of 50 µg ml⁻¹ and incubated with HEK293 stably expressing AOC3 with a transmembrane domain (i.e. located on the surface of the cell; figure 10b2). Comparing the wt forms showed that HEK293 cell-derived ZAG inhibited AOC3 activity, whereas the Expi293F cell-derived ZAG did not (figure 10c). Although the less-glycosylated form of Expi293F cell-derived ZAG is very likely to be present with the hyperglycosylated form, its inhibitory potential is strongly reduced. Hence, the inhibitory effect of recombinant ZAG depends on which expression host is used. Furthermore, the various ZAG glycomutants, produced in both HEK293 and Expi293F cells, showed a widely differing ability to inhibit AOC3. Importantly, the

loss of all glycosylation sites (Δ3-ZAG) led to the same inhibitory potential in both forms of the protein, whether produced in HEK293 or Expi293F cells, which confirms the impact of aberrant glycosylation. Since the ZAG molecular weight was not consistent in all plasma samples, plasma of different mouse strains was collected. WB analysis of mouse plasma of different mouse strains did not show a homogeneous pattern (figure 10d). Plasma ZAG of DBA and FVB mice showed a more disperse pattern, as observed when overexpressing ZAG in Expi293F cells.

## 3. Discussion

This work aimed to identify a new interaction partner of ZAG, which might help to explain its biological functions. Although the scientific literature is divided on the issue, most authors claim that ZAG acts via the β-adrenergic system.

The role of the adrenergic receptor system in ZAG-mediated lipolysis has been investigated *in vitro* using CHO-K1 cells, which were transfected with human $β_1$, $β_2$ and $β_3$ receptors [72]. The binding kinetics revealed that ZAG has an affinity for $β_2$ and $β_3$ receptors, but not for $β_1$ receptors. When transfected cells were incubated with recombinant human ZAG, there was an increase of cAMP levels that could be reduced by β-adrenergic antagonists [72]. Based on these *in vitro* results and the fact that ZAG deficiency leads to obesity [47], it was of interest to ask whether treatment with ZAG has anti-obesity and possibly anti-diabetic effects. Therefore, the *in vivo* effect of ZAG was studied in ob/ob mice, which are deficient in the hormone leptin and consequently suffer from obesity, hyperphagia and insulin resistance [73]. Studies performed with ob/ob mice showed that ZAG administration causes improved insulin sensitivity and reduced fat mass, which could be attenuated by the addition of the non-specific β-adrenoreceptor antagonist propranolol [69]. Another study with ZAG-treated male Wistar rats confirmed these results [74]. Therefore, it is very likely that ZAG acts as an adipokine and is directly involved in the breakdown of fat tissue.

However, another group directly compared the *in vivo* effects of recombinant ZAG and the $β_{3/2}$-agonist BRL35135 in ob/ob mice [75] and showed that, although there were similarities with previous published work, ZAG definitely did not behave as a $β_{3/2}$-agonist. Compared with the immediate effect of recognized $β_{3/2}$-agonists, the ZAG-mediated effect took several days. This correlates with the delayed lipolytic effect of ZAG compared with isoproterenol in 3T3-L1 cells (figure 7a). $β_{3/2}$-agonists also led to a downregulation of β-adrenoreceptors, which was not observed with ZAG [75].

Due to this inconsistency in how ZAG function is understood, a more direct approach was followed in this study to identify an interaction partner. The identification of AOC3 in this role highlights new possibilities for ZAG signalling and previously unsuspected functional relationships. To date, the only ligands known to interact with AOC3 are the sialic acid-binding immunoglobulin-type lectins Siglec-9 and Siglec-10 [76,77]. Interestingly, Siglec-10 serves as a substrate for AOC3, which deaminates an arginine residue [77]. AOC3 is an ectoenzyme and is strongly expressed on the surface of adipocytes and, during inflammation, on endothelial cells. On adipocytes, it comprises 2.3% of total plasma

membrane protein [78]. On endothelial cells, it promotes leucocyte adhesion and transmigration to sites of inflammation, which is not restricted to a specific immune cell population [44]. AOC3 is also shown to serve as homing receptor on high endothelial venules of lymphatic tissue, showing a strong binding preference for CD (cluster of differentiation) $8^+$-T-lymphocytes [79]. Homing of CD8$^+$-T-lymphocytes is necessary for antigen-specific priming and activation. It is noteworthy that antigen-specific priming and activation of CD8$^+$-T-lymphocytes during a viral infection leads to infection-associated-cachexia (IAC) [80]. In this regard it cannot be excluded that an interplay between AOC3 and ZAG could serve as additional co-regulator of priming and activation of T-lymphocytes in general or during IAC. Leucocyte transmigration in AOC3 k.o. mice is massively hampered, leading to abnormal leucocyte traffic [47] and strongly reduced leucocyte infiltration into adipose tissue [48]. AOC3 appears to have both catalytic and adhesive functions, although the molecular mechanism mediating leucocyte migration is incompletely understood. On the one hand, leucocyte adhesion is blocked by anti-AOC3 antibodies that do not inhibit enzyme activity. On the other hand, inactivation of the enzyme by a single point mutation—which is critical for enzyme activity—renders AOC3 unable to promote leucocyte migration [81]. AOC3 has also been shown to play a role in liver, lung and kidney fibrosis [82–84]. Treatment with the AOC3 inhibitor semicarbazide significantly reduced kidney fibrosis in a unilateral ureteric obstruction model in mice. Inhibition of AOC3 activity led to suppression of matrix gene expression, interstitial inflammation, oxidative stress and total collagen accumulation [84]. This matches the outcome in experimentally induced kidney fibrosis in ZAG k.o. mice [43]. ZAG deficiency leads to severe fibrosis, which can be rescued by injecting recombinant ZAG. If pharmacological inhibition of AOC3 blocks fibrosis, ZAG-dependent inhibition of AOC3 might produce a similar outcome [43]. Hence, AOC3 and ZAG are co-regulators for the development of fibrosis and ZAG-dependent inhibition of AOC3 might serve to attenuate this process.

Obesity is associated with adipose tissue inflammation and concomitant insulin resistance [85]. Obese patients have markedly reduced plasma concentrations of ZAG [86], which is explained by the elevated levels of TNF-α secreted by tissue-resident and activated macrophages [85]. Lean and healthy subjects have a higher plasma ZAG level and show no tissue inflammation [87]. From this perspective, it would be of interest to ask whether the reduced level of plasma ZAG observed in obese individuals results in reduced occupation of AOC3 on the surface of cells. If this were the case, more AOC3 molecules would be available for leucocyte adhesion and transmigration, which would promote insulin resistance. Enhanced plasma levels of ZAG might reduce inflammation-dependent transmigration and ameliorate its negative side effects, as already shown for chronically-administered AOC3 inhibitors [88]. Hence, it is tempting to speculate that increased or reduced levels of ZAG inversely correlate with the degree of inflammation observed in lean and obese people suffering from insulin resistance.

Similar logic could also explain the increased levels of ZAG observed in people suffering from cachexia. ZAG is one of the most prominent clinical markers of cachexia, which is highly upregulated during this energy-demanding state. However, inflammation of white adipose tissue is not

observed in patients suffering from cancer cachexia [89]. Nevertheless, unlike healthy controls and cancer patients not suffering weight loss, IL-6 plasma levels were strongly elevated [89], which fits the observation that ZAG expression is stimulated by hormones such as IL-8, leptin and IL-6 [90]. If ZAG is able to regulate leucocyte transmigration by binding to AOC3, elevated ZAG levels might act to prevent pronounced tissue inflammation and concomitant insulin resistance during cachexia.

The deamination of primary amines by AOC3 generates $H_2O_2$, which is known to activate insulin signalling [57]. Indeed, in AOC3-deficient mice, the stimulation of glucose uptake by AOC3 substrates is abolished, whereas insulin-stimulated glucose uptake remains unaffected [91]. Furthermore, acute and chronic administration of benzylamine increases glucose uptake in non-diabetic and diabetic rat models [92]. Inhibitors of AOC3 were also shown to have anti-obesity effects. Chronic administration of AOC3 inhibitors led to a reduced gain of fat adipose tissue in different mouse models on a high fat diet [93,94]. These findings support the indirect lipolytic effect of LJP1586 and ZAG by reduced deamination of lipolytic biogenic amines, as observed with octopamine in 3T3-L1 cells (figure 8a,b). However, this contradicts the observation that AOC3 k.o. mice have a significantly enlarged fat tissue mass compared with wt littermates [91]. In this regard, it should be noted that pharmacological inhibition does not always reflect a k.o. model [95] and undefined long-term counter-regulation of the nervous system cannot be excluded.

AOC3 substrates have been shown to inhibit lipolysis in isolated adipocytes [96], whereas ZAG is purported to stimulate lipolysis by binding to the $\beta_2$ and $\beta_3$ adrenoreceptors [72]. Using $H_2O_2$ as a signalling molecule, ZAG-mediated inhibition of AOC3 might serve as an alternative explanation of its lipolytic effect (figure 7). AOC3 substrates exert an insulin-like effect on adipocytes, and this is dependent on the formation of $H_2O_2$ [97]. $H_2O_2$ is a highly prevalent reactive oxygen species that controls enzyme activity by modulating the redox state of cysteine residues [98]. $H_2O_2$ is non-polar and able to diffuse through membranes or is transported through aquaporin 3 [99,100]. Although $H_2O_2$ is found throughout the cell, its signalling function is restricted and transduced by compartmentalization of antioxidant enzymes such as the peroxiredoxins [101]. Accordingly, AOC3-derived $H_2O_2$ could interfere with enzymes involved in stimulating lipolysis. Important components of this pathway are membrane-bound adenylate cyclase (AC), which generates cyclic adenosine monophosphate (cAMP), and the catalytic subunit of protein kinase A (PKA-C). Enhanced levels of cAMP bind to the regulatory unit of PKA, thereby releasing PKA-C, which in turn phosphorylates downstream elements, inducing lipolysis [61]. $H_2O_2$ increases levels of G($\alpha$)i—which reduces AC activity [102]—whereas PKA-C itself is inactivated by $H_2O_2$ [103,104]. Hence, binding of ZAG to AOC3 on adipocytes could potentially trigger lipolysis by reducing insulinogenic concentrations of $H_2O_2$ or by deamination of lipolytic biogenic amines, as observed in 3T3-L1 cells incubated with recombinant ZAG (figure 7a). AOC3/ZAG-dependent signalling could also involve trace amine-associated receptors (TAARs) [105], which form a subfamily of rhodopsin G-protein coupled receptors (GPCR). An important part of this signalling pathway is the heterotrimeric G-protein $G_s$, which is activated upon stimulation of GPCRs and promotes cAMP-dependent signalling by activating AC. Interestingly, $G_s$ is also associated with TAAR1. Therefore, indirect activation of TAAR1 due to higher concentrations of trace amines such as octopamine, which is released by platelets [106], or any other trace amine cannot be excluded. Notably, noradrenaline, serotonine, histamine and dopamine are also described as agonists of TAAR1 [107]. In this regard, two aspects are of interest. First, a physiological concentration of ZAG (50 µg ml$^{-1}$) shows almost the same inhibitory potential as the highly selective inhibitor LJP1586 (figure 5a,b). Second, a similar concentration of ZAG is sufficient to enhance octopamine-stimulated lipolysis in the low-micromolar range. Trace amine concentrations in plasma are also in the low- to sub-micromolar range [106,108]. Therefore, AOC3/ZAG-dependent changes in trace amine concentrations could strongly affect TAAR signalling.

Regarding octopamine-stimulated lipolysis in the presence of LJP1586 and ZAG (figure 8), it is notable that, compared with LJP1586, ZAG loses its stimulatory effect at higher octopamine concentrations. This difference in behaviour of ZAG and LJP1586 might reflect different types of inhibition. LJP1586 is a small molecule inhibitor that enters the catalytic site of the enzyme and is highly selective for AOC3 [30,109,110]. ZAG behaves like an allosteric inhibitor, i.e. it binds away from the active site, and reduces substrate affinity. On the one hand, AOC3-derived $H_2O_2$ is described as affecting its own enzyme activity [111,112]: the crystal structure of human AOC3 reveals a vicinal disulfide bridge [49], which is suggested to serve as a redox switch, possibly inducing a conformational change [113]. On the other hand, human ZAG contains one disulfide bridge in its MHC-fold and one inter-sheet disulfide bridge in the immunoglobulin (Ig)-like domain. Oxidation of disulfides by $H_2O_2$ and one- or two-electron oxidants at physiological pH results in the formation of disulfide monoxides or disulfide dioxides, which further leads to cleavage of disulfides and the formation of sulfonic acid [114–116]. Notably, copper-containing amine oxidases also form hydroxyl radicals (one-electron oxidants) due to the reaction between $H_2O_2$ and reduced copper [117]. Extracellular proteins mainly contain disulfides [118] and are exposed to a higher level of ROS in general [119]. Modification of the disulfides in receptors and plasma proteins is involved in protein stability [120], protein oligomerization [121], the transformation of biological function [122,123] and receptor–ligand interaction [124]. A similar interplay between ROS signalling, ligand recognition and protein–protein interaction is imaginable for AOC3 and ZAG, which could restrict the inhibitory function of ZAG. To test whether AOC3 activity affects protein–protein interaction, wt and different glycoforms of recombinant ZAG were incubated with benzylamine in the presence or absence of recombinant AOC3 (electronic supplementary material, figure S3). In the presence of AOC3, only wt ZAG shifts to a higher molecular weight, whereas in the absence of AOC3 it does not. By contrast, incubation with $H_2O_2$ induces a shift in wt ZAG, irrespective of whether AOC3 is present. This could hint at an oxidation-dependent oligomerization of ZAG, influencing AOC3/ZAG and/or ZAG/ZAG protein–protein interaction, in which glycosylation plays an additional role. ZAG oligomerization could serve as a self-regulatory mechanism, and explain why a complete inhibition of activity was never observed at an equimolar ratio of both proteins (figure 4c), as well as why ZAG loses its lipolysis-stimulatory effect (figure 8).

Besides $H_2O_2$, $NH_3$ might also serve as a signalling molecule. Compared with $H_2O_2$, less is known about its

function in this context. $NH_3$ is known to stimulate autophagy, playing an important role in energy metabolism in tumour cells [125]. In summary, $H_2O_2$ and perhaps also $NH_3$ may have currently uncharacterized effects on AOC3 activity—with or without ZAG modulation—that interfere with signalling pathways. This represents a challenge to researchers to identify physiological compounds serving as substrates for AOC3.

During this study, many different expression hosts were tested to find a way to express both AOC3 and ZAG in sufficient, biologically active quantities. Specifically, a surprising difference between HEK293 and Expi293F was observed. Both derive from the same attached cell line, but the latter has been adapted to grow in suspension. Compared with HEK293 cells, overexpression of ZAG in Expi293F cells results in a hyperglycosylated and—to a lesser extent—hypoglycosylated form (figure 10a2). Different glycosylated forms of ZAG were previously identified by isoelectric focusing and are found in plasma, amniotic fluid, saliva and tears [64]. The carbohydrate content of human plasma-derived ZAG makes up to 12–15% of total mass [65]. By contrast, human seminal fluid-derived ZAG contains no carbohydrate [126]. One publication analysed murine ZAG of plasma and different tissues by WB. Interestingly, ZAG had different molecular weights in most tissues and plasma [47]. This is in line with the observation that plasma ZAG from different mouse strains also shows no coherent pattern (figure 10d). Tunicamycin, BAGN and PNGase F treatment of purified ZAG proteins confirmed that size differences originate from N-glycosylation. Strikingly, the addition of 500 µM $CoCl_2$—a hypoxia mimetic that stabilizes the transcription factor HIF1-α—reverses this effect (figure 10a3,4). Glycosylation of proteins is highly variable among individuals and is influenced by oxygen levels. For instance, hypoxia has been shown to reduce uridine diphosphate N-acetyl-glucosamine (UDP-GlcNAC) levels [127]. This fact is explained by HIF1-α-induced transcription of pyruvate dehydrogenase kinase (PDK) and inactivation of pyruvate dehydrogenase by PDK. As a result, production of acetyl-CoA (coenzyme A) is suppressed, such that acetylation of glucosamine and biosynthesis of UDP-GlcNAC are reduced [128]. Hypoxia also limits production of nucleotides such as ATP, GTP, UTP and CTP, which might also interfere with the addition of UDP to GlcNAC [129]. Hence, a higher oxygen level causes the opposite effects, as observed with Expi293F cells. Differences in the carbohydrate content of plasma ZAG and seminal fluid ZAG are thought to affect physiological function [126]. The interdependence of glycosylation and physiological function has been described for many other proteins [50]. For example, glycoproteomic profiling of glycodelin revealed different isoforms, each of which contains unique carbohydrates associated with different functions involved in capacitation, acrosome reaction, immune suppression or apoptosis [130–134]. This finding might support the idea that hyperglycosylated ZAG, which is produced when overexpressed in Expi293F cells and shows markedly reduced inhibition of AOC3 activity (figure 10c), corresponds to one specific in vivo glycoform and thus might have a particular physiological impact. Modification of the carbohydrate content of recombinant ZAG in the presence of $CoCl_2$ provokes the notion that differences in ZAG glycosylation are co-regulated by oxygen-sensing factors and that these differences affect biological function in vivo. Diseases associated with a rise in ZAG levels, such as cancer, AIDS [37] and chronic heart and kidney disease [40,135,136], often manifest dyspnoea due to highly interdependent symptoms such as fatigue, physical impairment, pulmonary hypertension, lung infections and heart failure [137–139]. It would be interesting to observe whether overall oxygen saturation affects the glycosylation pattern of ZAG. Moreover, a paraneoplastic syndrome such as cachexia has also been attributed to ZAG secretion by tumour cells, which contributes to a rise in plasma ZAG levels [7]. Since hypoxia is a characteristic feature of solid tumours, it cannot be excluded that this also affects the glycosylation and biological function of ZAG secreted by tumour cells [140,141]. Clinical studies on ZAG have been solely based on the quantification of ZAG by qRT-PCR, ELISA or tissue microarray-based immunohistochemistry. However, the amount of ZAG might not be as important as the form of its glycosylation. Glycoproteomic profiling or at least precise estimation of its molecular weight might offer deeper insights into the true biological function of ZAG. Taken together, the recognition of ZAG as an allosteric inhibitor of ectoenzyme AOC3 should prompt a reinterpretation of ZAG-associated functions, in particular its pro-lipolytic and anti-inflammatory roles.

# 4. Methods

## 4.1. Protein expression and purification

E. coli BL21 (DE3): Both human and murine ZAG (without leader sequence) were produced in E. coli after cloning in the expression plasmid pGEX-6P-2 which adds an N-terminal GST -tag to each recombinant protein, enabling affinity purification using glutathione (GSH)-Sepharose. GST-tag alone was purified from E. coli by using pGEX-6P-2 as expression plasmid without gene of interest. Murine ZAG (mZAG fw (XmaI): GCCC 5′GGGGTGCCTGTCCTGCTGTC; mZAG rev (XhoI): 5′GCTCGAGTTACTGAGGCTGAGCTACAA) and human ZAG (hZAG fw (XmaI): 5′TCCC-GGGGTAAGAATGGTGCCTGTCCT; hZAG rev (XhoI): 5′ TCTCGAGCTAGCTGGCCTCCCAGGGCA) were amplified by PCR from cDNA of murine liver and HEPG2 cells (ATCC Cat# HB-8065, RRID:CVCL_0027), respectively. E. coli cells (carrying the expression plasmid pGEX-6P-2-hZAG or pGEX-6P-2-mZAG) from glycerol stocks were freshly streaked on LB medium agar plates containing the appropriate selection marker. For GST-ZAG overexpression, a 5 ml overnight culture was set up. The following day, 3 ml of the overnight culture was inoculated into 300 ml LB medium and grown to an $OD_{600}$ of around 0.8–1.0. The temperature was reduced to 25°C and cells were induced with 50 µM IPTG for 3 h. For the isolation of recombinant GST-ZAG, cells were harvested at 4000g and 4°C for 10 min and resuspended in $1 \times$ PBS supplemented with 10 mM EDTA and lysozyme (100 µg $ml^{-1}$). The suspension was incubated on a rocking plate for 30 min and then frozen at −80°C. The frozen suspension was thawed in a water bath at 37°C. The viscous cell suspension was supplemented with 10 mM $MgSO_4$ and 10 units DNAse (Roche) and incubated for 5–10 min at 37°C in a water bath. Subsequently, the lysate was centrifuged at 15 000g and 4°C for 20 min and incubated with 400 µl pre-equilibrated glutathione (GSH)-Sepharose on an over-the-top wheel. GSH-Sepharose was collected by

royalsocietypublishing.org/journal/rsob    Open Biol. 10: 190035

royalsocietypublishing.org/journal/rsob Open Biol. **10**: 190035

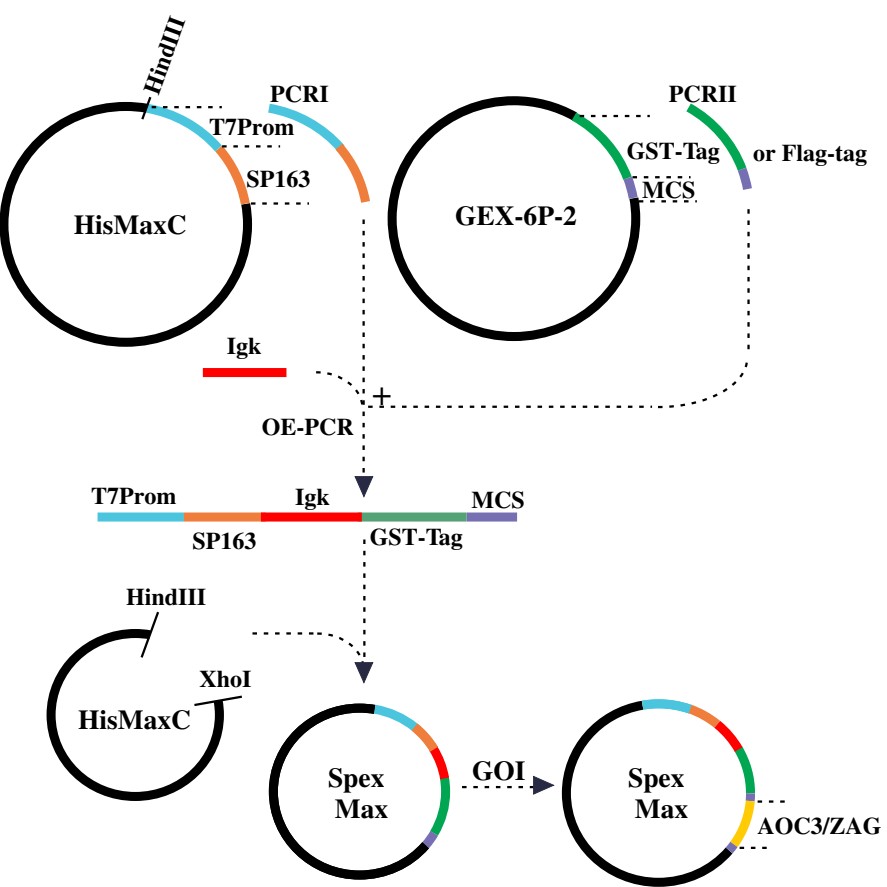

**Figure 11.** Construction of expression plasmid pSpexMax: the pSpexMax expression plasmid is largely a combination of pcDNA4/HisMax C and pGEX-6P-2. Partial sequences of pcDNA4/HisMax C (T7 promotor and SP163 translational enhancer sequence) and pGEX-6P-2 (GST-tag, including cleavage site, and multiple cloning site (MCS)) and the leader sequence of Ig kappa light chain were amplified by PCR and ligated by overlap-extension-PCR (OE-PCR). The PCR products and pcDNA4/HisMax C were digested with HindIII/XhoI and ligated, resulting in pSpexMax. The coding sequences of AOC3 and ZAG (GOI, gene of interest) were cloned into the expression plasmid and tested for expression.

centrifugation and washed with 1 × PBS. Protein was eluted with 10 mM reduced GSH (Sigma) dissolved in 10 mM Tris-HCl, 150 mM NaCl, pH 8 and dialysed against 1 × PBS.

## 4.2. Cell culture

*3T3-L1 Cells* (ATCC Cat# CL-173, RRID:CVCL_0123): Before seeding, multi-well plates were coated with 0.2% gelatin and left overnight. Cells were grown in DMEM high glucose (Gibco) supplemented with 10% FCS (Gibco) until 2 days after becoming confluent. To stimulate differentiation, the medium was supplemented with 4 µg ml$^{-1}$ dexamethasone, 10 µg ml$^{-1}$ insulin and 500 µM isobutymethylxanthine (IBMX). After 3 days, the medium was replaced with medium supplemented only with insulin (10 µg ml$^{-1}$), which was changed every second day. After four more days, the insulin concentration was further reduced to a final concentration of 0.2 µg ml$^{-1}$ and left until lipid droplets developed. The medium was changed every third day.

*SGBS cells* (RRID:CVCL_GS28): For differentiation, the following media were prepared: 0F (DMEM F-12, 1% Biotin, 1% pantothenic acid, 1% penicillin/streptomycin, 10% FCS), 3FCB Dex/Mix (DMEM F-12, 1% Biotin, 1% pantothenic acid, 1% penicillin/streptomycin, 0.01 mg ml$^{-1}$ transferrin, 0.1 µM cortisol, 200 pM tri-iodothyronine, 20 nM human insulin, 0.25 µM dexamethasone, 500 µM IBMX, 2 µM rosiglitazone), 3FC Dex/Mix (3FCB Dex/Mix without rosiglitazone) and 3FC (3FC Dex/Mix without

dexamethasone and IBMX). 0F medium was used for cultivating SGBS cells and changed twice per week. For differentiation, $2 \times 10^5$ cells per 10 cm culture dish were seeded and grown until confluency. The growth medium was switched to 3FCB Dex/Mix for 3 days and changed to 3FC Dex/Mix on the fourth day. On the seventh and eleventh day, the medium was replaced with 3FC. Lipid droplets developed after two weeks of differentiation.

*HCAECs*: HCAECs (kindly provided by Gunther Marsche) were cultured in six-well plates coated with 1% gelatin and left overnight. Cells were grown and used for experiments until they reached the ninth passage. Special medium was provided by Lonza (EGM™-2 MV Microvascular Endothelial Cell Growth Medium mixed with supplements according to manufacturer's protocol: hydrocortisone, hFGF-B, VEGF, R3-IGF-1, ascorbic acid, hEGF, and GA-1000).

*HEK293 cells*: HEK293 cells (CLS Cat# 300192/p777_HEK293, RRID:CVCL_0045) were cultured in DMEM high glucose (Gibco) and supplemented with 10% FCS (Gibco). Cells were split on reaching 80% confluency.

*Expi293F cells*: Expi293F cells (RRID:CVCL_D615, kindly provided by Walter Keller) were cultivated in a ventilated 125 ml disposable shaker flask (Corning) and maintained on an orbital shaker. Cells were grown in Expi293 Expression Medium (Gibco) and split 1 : 10 on reaching a density of $5 \times 10^6$ cells ml$^{-1}$.

All cells were grown in a $CO_2$-controlled incubator with a relative humidity of 90% at 37°C.

royalsocietypublishing.org/journal/rsob    Open Biol. **10**: 190035

## 4.3. Construction of expression plasmid pSpexMax

An expression plasmid, pSpexMax, was constructed as shown in figure 11 for the production of both murine AOC3 and murine ZAG in mammalian cells. The leader sequence of Ig kappa light chain was taken from Ohman et al. [48], which directs the protein into the medium. The SP163 translational enhancer sequence was incorporated upstream of the leader sequence to promote recombinant protein translation, while the GST-tag, equipped with a cleavage site (recognized by PreScission Protease; GE Healthcare), was inserted downstream of the leader sequence to facilitate affinity purification. For transient expression, GST-tag alone was purified from Expi293F cells by using pSpexMax as expression plasmid without gene of interest. For large-scale purification of both proteins, the whole sequence (SP163, Igk, GST and AOC3/ZAG) was amplified by PCR and cloned into the expression plasmid pLVX-Tight Puro (Clontech), which allows packaging of constructs in a lentiviral format. For GST purification from stable HEK293 cells only SP163, Igk and GST were PCR amplified and cloned into the expression plasmid pLVX-Tight-Puro. Lentivirus versions of pTET-off and pLVX-Tight Puro (AOC3/ZAG) were used to transduce HEK293 cells.

*Production of lentivirus*: For production of lentivirus, Lenti-X™, the HTX Packaging System (Clontech), was used following the manufacturer's protocol. Briefly, murine AOC3 and murine ZAG were cloned into the plasmid pLVX-Tight-Puro. For virus production, $5 \times 10^6$ HEK293T cells were seeded in a 10 cm dish 24 h before transfection. The Xfect Transfection System (Clontech) was used for transfection of lentiviral plasmids, pLVX-Tight Puro and pTET-Off (Clontech). After 2 days, the virus-containing medium was collected and centrifuged at $1200g$ for 2 min. Supernatant was aliquoted and stored at $-80°C$.

*Transduction and selection of HEK293 cells*: Twenty-four hours before transduction, a six-well plate was seeded with $3 \times 10^5$ HEK293 cells per well. On the day of transduction, medium was supplemented with 8 µg ml$^{-1}$ hexadimethrine bromide (Sigma) and virus. Plates were centrifuged at $1200g$ and $32°C$ for 1 h and incubated for another 24 h. Subsequently, the medium was replaced with a medium containing both selection markers, puromycin (2 µg ml$^{-1}$) and G-418 (400 µg ml$^{-1}$). After selection, conditioned medium and stable cells were analysed for protein expression by WB.

*Lentivirally transduced HEK293 cells*: Lentivirally transduced HEK293 cells, stably secreting GST-AOC3 and GST-ZAG, were grown until they became confluent. On every third day, conditioned medium was collected and stored at $-20°C$. For protein isolation, 500 ml frozen medium was thawed and incubated with 200 µl GSH-Sepharose. Subsequently, the protein was eluted with 10 mM reduced GSH (Sigma) in 10 mM Tris-HCl, pH 8 and 150 mM NaCl. The purified protein was dialysed against $1 \times$ PBS, the GST-tag was removed by PreScission Protease and the released protein dialysed against $1 \times$ PBS. Protein integrity was checked by SDS-PAGE.

*Expi293F cells*: Transfections were performed using the ExpiFectamine 293 Transfection Kit (Gibco) following the manufacturer's protocol. Briefly, cells were diluted to a density of $3 \times 10^6$ cell ml$^{-1}$ with a fresh medium. Plasmid DNA (1 µg ml$^{-1}$ culture) and ExpiFectamine 293 Reagent (Gibco) were diluted in Opti-MEM I medium (Gibco) and mixed by inverting. After 10 min, the reaction was added to suspension cultures. After 18 h, the enhancer solutions ExpiFectamine 293 Transfection Enhancer 1 and ExpiFectamine 293 Transfection Enhancer 2 (Gibco) were added. After 72 or 96 h, the medium was collected and prepared for GST affinity purification or WB. Overexpression experiments were performed in 125 ml disposable shaker flasks or six-well plates.

## 4.4. Amine oxidase assays

*AOC3 activity measurement using Amplex Red*: For fluorescent measurement of amine oxidase activity, AOC3 standard reagent Amplex Red (Invitrogen) was used. The signal was measured at an excitation/emission ratio of 560/590 nm. All measurements were performed at $37°C$ by connecting the fluorimeter (DU 640 Spektrometer, Beckman) to a water bath. Only sterile-filtered $1 \times$ PBS was used as a reaction buffer since autoclaving produced non-defined peroxide species, which caused false positive signals. The standard reaction (500 µl) comprised 50 ng AOC3, 4 µM Amplex Red and two units HRPO (Sigma). For inhibition, the sample was pre-incubated with LJP1586 (La Jolla Pharmaceuticals) or ZAG for 5 min at $37°C$. The reaction was started by addition of 5 µl 10 mM benzylamine (Sigma) and stopped by adding 10 µl Amplex™ Red/UltraRed Stop Reagent (Invitrogen).

*Radioactive AOC3 assays*: The standard reaction (500 µl) comprised 50 ng AOC3, 100 µM benzylamine, 1 Ci/mol [$^{14}$C]-benzylamine (PerkinElmer) and $1 \times$ PBS. For inhibition, the sample was pre-incubated with LJP1586 or ZAG for 5 min at $37°C$. The sample was incubated at $37°C$ in a water bath for 60 min. After incubation, the reaction was stopped by adding 20 µl of 2 M HCl per 100 µl reaction volume followed by 200 µl of extraction solvent (toluene/ethyl acetate, 1 : 1, v/v) per 100 µl reaction volume. Samples were vortexed and centrifuged at $700g$ for 10 min, then 200 µl of the upper organic phase (approx. 850 µl) were measured by liquid scintillation counting.

*Radioactive cell culture experiment*: The day before the experiment, stable HEK293 were seeded at a density of $2.5 \times 10^5$ cells well$^{-1}$ (6-well plate). 3T3-L1 adipocytes were used when fully differentiated and HCAEC cells when confluent. For measurement, the cells were incubated with the corresponding media without FCS supplemented with 100 µM benzylamine and 1 Ci/mol [$^{14}$C]-benzylamine. For inhibition, cells were pre-incubated with LJP1586 or ZAG for 15 min. After 30 min incubation, supernatant was collected, extracted and measured according to the standard radioactive AOC3 assay. Cells were washed three times with 1 ml $1 \times$ PBS and lysed by incubation with 0.3 M NaOH/1% SDS. Protein amount was quantitated using BCA reagent.

*AOC3 activity measurement using 4-nitrophenylboronic acid pinacol ester (NPBE)*: This is a colorimetric assay based on the oxidation of NPBE in the presence of $H_2O_2$ [59]. The standard reaction (250 µl) comprised 50 mM potassium phosphate buffer pH 7.4, 10 µg AOC3, 150 mM NaCl, 100 µM NPBE (ethanolic solution) and 20 mM substrate. Samples were incubated at $37°C$ and stopped by adding 1 mM DTT and 5 µl 5 M NaOH.

royalsocietypublishing.org/journal/rsob    Open Biol. 10: 190035

## 4.5. Cross-linking

Plasma membrane was isolated according to Belsham *et al.* [142]. The gonadal adipose tissue of 10 C57Bl6 mice (older than four months) or differentiated SGBS cells ($5 \times 10$ cm culture dishes) were collected and mixed with 1 ml of sucrose-based medium (SBM1) (10 mM Tris-HCl, 0.25 M sucrose, 80 mM EGTA, pH 8.2) and homogenized on ice. Samples were centrifuged for 30 s at $1000g$, the infranatant was collected with a syringe, pooled and centrifuged at 4°C at $30\,000g$ for 30 min. The pellet was resuspended in 500 µl SBM1. Two tubes were filled with 8 ml 'self-forming gradient of percoll' comprising Percoll (80 mM Tris-HCl pH 8, 2 M sucrose, 80 mM EGTA), SBM2 (10 mM Tris-HCl pH 8, 0.25 M sucrose, 2 mM EGTA) and SBM1, mixed in a ratio of 7 : 1 : 32 (v/v/v). The resuspended pellet was gently loaded onto the gradient solution and centrifuged at 4°C and $10\,000g$ for 15 min. After centrifugation, a fluffy white band at the bottom was collected with a large gauge needle, washed two times with $1 \times$ PBS and pelleted by centrifugation at $10\,000g$. The pellet was finally resuspended in 500 µl 0.25 M sucrose dissolved in $1 \times$ PBS.

Purified proteins were labelled with Sulfo-SBED (Thermo Scientific) according to the manufacturer's protocol (figure 1). Sulfo-SBED comprises biotin, a sulfated N-hydroxysuccinimide (Sulfo-NHS) active ester and a photoactivatable aryl-azide. Successful labelling of human or mouse GST-ZAG and GST was confirmed by WB, and labelled proteins were extensively dialysed against $1 \times$ PBS to eliminate any non-bound Sulfo-SBED molecules. In a dark room (with red safety light), 100 µg of labelled proteins were mixed with 100 µl of freshly-isolated membranes in a six-well plate and then wells were filled to a final volume of 500 µl with $1 \times$ PBS. Plates were wrapped in aluminium foil and incubated on a rocking plate at 4°C for 1 h. Subsequently, samples were exposed to UV light while cooling on ice. The protein solutions were transferred to a 1.5 ml tube and delipidated by addition of 0.5% N-octyl-glucoside. Delipidated proteins were either directly separated by SDS-PAGE or incubated with 50 µl of streptavidin agarose. Agarose-bound proteins were washed four times with $1 \times$ PBS, once with 0.5 M NaCl and then eluted with a $1 \times$ SDS loading buffer. Eluted samples were separated by SDS-PAGE. After SDS-PAGE, samples were either analysed by WB or Coomassie Brilliant Blue-stained bands were excised with a scalpel and subjected to LC-MS/MS.

## 4.6. Peptide sequencing by LC-MS/MS

Excised gel bands were washed with 150 µl distilled $H_2O$, 150 µl 50% acetonitrile and 150 µl 100% acetonitrile with a brief centrifugation step in-between. After the last washing step samples were dried in a vacuum centrifuge. Dehydrated samples were reduced by adding 60 µl 10 mM DTT dissolved in 100 mM $NH_4HCO_3$ and incubated at 56°C for 1 h. After cooling, the supernatant was removed and replaced with 55 mM 2-iodoacetamide dissolved in 100 mM $NH_4HCO_3$. After 1 h incubation, samples were washed with 100 mM $NH_4HCO_3$ and then dehydrated and swollen by adding 50% acetonitrile and 100 mM $NH_4HCO_3$, respectively. Treated samples were dried in a vacuum centrifuge. Subsequently, gel pieces were swollen by a stepwise addition of digestion buffer (50 mM $NH_4HCO_3$, 5 mM $CaCl_2$, and 12.5 ng $µl^{-1}$

trypsin) on ice. Samples were covered with a digestion buffer and incubated at 37°C in a thermomixer overnight. The next day, peptides were extracted by adding 35 µl 1% formic acid and 160 µl 2% acetonitrile followed by 35 µl 0.5% formic acid and 160 µl 50% acetonitrile. Supernatants were collected and dried in a vacuum centrifuge. Extracted proteins were resuspended in 0.1% formic acid separated on a nano-HPLC system (Ultimate 3000, LC Packings, Amsterdam, Netherlands), with a flow rate of 20 µl $min^{-1}$ using 0.1% formic acid as a mobile phase. Loaded samples were transferred to a nano-column (LC Packings C18 PepMap, 75 µm inner diameter $\times$ 150 mm) and eluted with a flow rate of 300 nl $min^{-1}$ (solvent A: 0.3% aqueous formic acid solution; solvent B: water/acetonitrile 20/80 (v/v), 0.3% formic acid; gradient: 5 min 4% solvent B, 35 min 55% solvent B, 5 min 90% solvent B, 47 min 4% solvent B). Samples were ionized by a Finnigan nano-ESI source, equipped with NanoSpray tips (PicoTip Emitter, New Objective, Woburn, MA). Analysis was performed by Thermo-Finnigan LTQ linear ion-trap mass-spectrometer (Thermo, San Jose, CA, USA). MS/MS data were synchronized with the NCBI (26.9.2010) non-redundant public database with SpectrumMill Rev. 03.03.084 SR4 (Agilent, Darmstadt, GER) software. For identification, at least three or more different peptide sequences must be detected [143].

## 4.7. Ion exchange chromatography

Four overnight-fasted C57Bl6 mice were anaesthetized using isoflurane and blood was collected via the retro-orbital sinus. Protein from 2 ml collected murine plasma was desalted and rebuffered in 10 mM Tris-HCl, pH 8 using a PD-10 desalting column (GE Healthcare) and further diluted to a final volume of 20 ml using 10 mM Tris-HCl, pH 8. Plasma proteins were separated by anion exchange chromatography using Resource Q column (GE Healthcare, 6 ml) connected to an ÄKTA Avant 25 system (GE Healthcare). After loading, the column was washed with 10 column volumes of binding buffer and bound proteins were eluted by linear salt gradient (0–1 M NaCl). The protein concentration of all fractions was measured and ZAG-containing fractions identified by WB.

## 4.8. GST pulldown

GST-tagged AOC3 isolated from the conditioned medium of lentivirally transduced HEK293 cells was incubated with 1 : 10 diluted murine plasma. The reaction comprised 100 µl diluted murine plasma, 200 µl recombinant GST-AOC3 (50 µg $ml^{-1}$), 50 µl pre-equilibrated GSH-sepharose and 150 µl $1 \times$ PBS. The reaction was incubated on an over-the-top wheel, centrifuged at $700g$ for 1 min and then the flow-through was collected. The GSH-sepharose was washed five times with 500 µl $1 \times$ PBS and bound proteins were eluted with $1 \times$ SDS loading buffer. Samples were analysed by WB.

## 4.9. Glycerol measurement

The medium of stimulated 3T3-L1 adipocytes was collected and glycerol content measured using a standard glycerol kit (Sigma). Cells were washed three times with $1 \times$ PBS and lysed by incubation with 0.3 M NaOH and 1% SDS. Protein was quantitated using BCA reagent (Pierce).

## 4.10. Western blot

Proteins were separated by 10% SDS-PAGE according to standard protocols and blotted onto polyvinylidene fluoride membrane (Carl Roth GmbH). Membranes were blocked with 10% blotting grade milk powder (Roth) in TST (50 mM Tris-HCl, 150 mM NaCl, 0.1% Tween-20, pH 7.4) at room temperature for 1 h or at 4°C overnight. Primary antibodies were directed against GST (GE Healthcare Cat# 27-4577-01, RRID:AB_771432), murine AOC3 (Abcam Cat# ab42885, RRID:AB_946102) and ZAG (Santa Cruz Biotechnology catalogue no. sc-11245, RRID:AB_2290216). Signals were visualized by enhanced chemiluminescence detection (Clarity Western ECL Substrate, Bio-Rad) and the ChemiDoc Touch Imaging System (Bio-Rad).

## 4.11. Statistical analysis

Statistical analysis and diagrams were prepared using Graph-Pad Prism 8.0.1 (GraphPad Prism, RRID:SCR_002798).

Figures and illustrations were prepared using CorelDRAW 2018 (CorelDRAW Graphics Suite, RRID:SCR_014235).

Data accessibility. This article has no additional data.

Competing interests. I declare I have no competing interests.

Funding. This study was supported by Austrian Science Fund (TRP 4 Translational Research Programme, P 26166 and P 28286) and Medizinische Universität Wien (Hans und Blanca Moser Stiftung).

Acknowledgements. I am deeply grateful to Gerhard Hofer (Karl-Franzens-University, Graz) for helping with cell culture experiments, discussions and proofreading the manuscript, and Roland Viertlmayr (GE-Healthcare, Munich) for helping with ion exchange chromatography and his invaluable expertise for substantially improving experiments. I want to thank Friedrich Buck of the core facility of mass-spectrometry based proteomics, UKE (Hamburg), whose technical advice contributed decisively to successful peptide sequencing. I want to thank Gerald Rechberger (Karl-Franzens-University, Graz) for performing LC-MS/MS based peptide sequencing. Finally, I would like to thank Walter Keller (Karl-Franzens-University, Graz) for providing Expi293F cells and Gunther Marsche (Medical University, Graz) for providing HCAEC cells.

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
