## [Reviewer comments · Open Biology]

Review History

RSOB-19-0035.R0 (Original submission)

Review form: Reviewer 1

Recommendation

Major revision is needed (please make suggestions in comments)

Are each of the following suitable for general readers?

- a) **Title**
Yes
- b) **Summary**
Yes
- c) **Introduction**
Yes

Is the length of the paper justified?

Yes

Should the paper be seen by a specialist statistical reviewer?

No

Is it clear how to make all supporting data available?

No

Is the supplementary material necessary; and if so is it adequate and clear?

No

Do you have any ethical concerns with this paper?

No

Comments to the Author

The manuscript is dealing with the biological functions of a soluble protein known from a while to be related with cachexia, therefore with energy balance, the Zinc-alpha2-glycoprotein (ZAG). Reported data show binding of ZAG to the membrane form of Vascular adhesion protein-1 (VAP-1), an adhesion molecule endowed with a semicarbazide-sensitive amine oxidase (SSAO) activity. SSAO/VAP-1 is abundant in adipocytes, vessels, and sites of inflammation, and supposed to generate reactive oxygen species once oxidizing amines. The interaction between these two proteins leads to the inhibition of SSAO activity. Thereby, several biogenic amines being substrates of SSAO/VAP-1, have likely their half-life increased and can longer stimulate their receptors. This seems to be the case of octopamine, which activates beta3-adrenoceptors and therefore increases lipolysis in adipocytes, probably making the link between ZAG and beta3-adrenoceptor activation proposed by various previous reports. The present findings support the conclusions of the author and surprisingly suggest that two endogenous multifunctional proteins might have uncovered incidences on diabetes and obesity. However, such novel important message deserves a better presentation of the manuscript, which suffers from numerous minor defects, and which needs in-deep revision, into the opinion of reviewer, alongside three other concerns.

A) As far as obesity or diabetes are concerned, a major concern could be related about the absence of clinically relevant observations, but the message being so novel, the preclinical observations are largely sufficient for publication in open Biology.

B) Another concern is that taking into account the variety of methods used in this report, it looks strange to the reviewer that the submitted work, which is not a review, is signed by only one author?

c) Then, the English style mixing past and present, and using commas too scarcely, has to be improved substantially; as it is the case of the last sentence of introduction ... and the first of conclusion. Moreover, the manuscript has not numbered lines, which did not facilitate building the necessary list of amendments, which is given below in a non-exhaustive manner:

- 1) check spelling of concavalin A in p2, first paragraph.
- 2) is ref 44 really dealing with the interplay between ZAG and β 3-adrenoceptors?
- 3) the abbreviations GST and PNGase are not introduced in any part of the Manuscript (Ms). Moreover GST-tag is even less described
- 4) Fig 1 is unreadable, at least at the reduced size chosen for the manuscript: For instance the ZAG indicated in a yellow circle is much lower than Times/Helvetica 8. Moreover, this figure has nothing to do in the Result section, and likely will be better located in Methods.
- 5) Figs 2 and 3 also contain panels that are too much reduced in their final appearance. The reviewer needed a X300 enlargement to distinguish items in several panels.
- 6) errors in title of fig 3. Can author(s) replace(s) by: ZAG modifies AOC3 activity ?

- For panels B , D and related, the units of Y-axis are not correct , they seem to be in pmol H₂O₂ /mg protein/min in reality ? More precisely, authors have to consider that activity is not normalized per mg of proteins as proposed , but per mg of AOC3 recombinant protein, since when adding ZAG to AOC3 protein , the amount of protein is increasing in the well and subsequently the proportion of active amine oxidase is decreasing, therefore leading to a lowered velocity per mg of (total) proteins , which should be observable with the addition of any other protein, even non-reacting with SSAO activity . Otherwise, it must be stated that the same amount of enzyme is distributed in control and assays in the presence of increasing doses of ZAG. Also applies for endogenous SSAO activity of 3T3 and coronary cells, for Fig 4B , and for supplemental data. Authors have to indicate in X-axes that they tested 12.5 ng ZAG and not 12,5.
- 7) Chemical name of LJP1586 has to be provided and source indicated in Methods.
 - 8) Fig 4B the increase of SSAO activity by plasma fractions devoid of ZAG is even greater than the inhibition by the soluble protein . Can the authors add other tests on this effect rather than building an hypothesis on stimulatory effects of albumin to state more unambiguously on this issue? Supplemental data needs much comments, especially the repeated lacks of dose-response.
 - 9) Fig 5 also suffers of too much numerous and reduced/small panels. "Noradrenaline" must be corrected for each occurrence/. As the same research experiment was performed in panels D and G, reviewer proposes to separate panels G and H as a novel figure. This will show with insistence that G and H were (perhaps?) performed in parallel and will authorize the authors to confirm their findings originally observed in 5D in a further approach that consisted in comparing a SSAO inhibitor and ZAG. Otherwise, remove these two panels and make a fusion of their common observations, since for scientific publications, a rule is not to show repeatedly the same results of a same experiment, save if normalization of data has an influence on interpretation...
 - 10) Obviously, the lack of inhibition by plasma from ZAG ko mice indicates in a clearer manner what component of plasma as involved in the inhibition of AOC3. Why not testing addition of BSA to ZAG ko plasma?
 - 11) Conclusion/interpretation of the data are given at the end of the figure legends, whilst this is generally indicated in Results section referring to the figure. Thus, most of the last sentences of the figure legends have to be reinserted in Results or deleted when already present.
 - 12) The simple observational statement at the end of p10 needs further attention: Authors should comment why ZAG is not behaving as the soluble SSAO inhibitor regarding the improvement of lipolysis in response to low or high doses of octopamine?
 - 13) Regarding the influence of ZAG glycosylation state on SSAO inhibition, the test of human plasma would have brought more clinically relevant info than the comparison of multiple forms with different glycosylation patterns from diverse sources, even including human embryo. Facing to a so complex approach of various MW forms of ZAG, the bullet sentence " The inhibitory effect clearly depends on the expression host " in p 13 is not so informative as it appears and needs further clarification. On the other hand, the first sentence of discussion is revealing an objective that was not declared before and that does not seem to be supported by provided data. Thus, the search for a "ZAG receptor" should be presented with more caution or replaced by the term "interaction partner" more adequately used in the discussion.
 - 14) The passage in discussion about hydrogen peroxide-dependent signalling is too long since not any experiment, except the measurement of its production during amine oxidation, has been performed in this way.
 - 15) many typo errors have to be checked throughout the Ms.
 - 16) Lastly, not all the doi are provided in the reference list.

Review form: Reviewer 2

Recommendation

Accept with minor revision (please list in comments)

Comments to the Author

Zinc-alpha2-glycoprotein (ZAG) plasma protein is an important biomarker for the diagnosis of cachexia. ZAG is known to cause progressive weight loss by increasing β -adrenergic receptor-mediated lipolysis in adipocytes. In this manuscript, the authors isolated amine oxidase (AOC)3 as a ZAG interacting protein. AOC3 is a copper-containing cell surface protein, expressed mainly in adipocytes and endothelial cells, that can affect both lipolysis and extravasation of immune cells. Further, they showed that ZAG modulates AOC3-dependent lipolysis in adipocytes and leukocyte transmigration by acting as an allosteric inhibitor of AOC3. Based on these results, the authors concluded that ZAG also acts on lipolysis in a β -adrenergic signalling independent manner. The findings are interesting but addressing the following comments will strengthen the manuscript.

Comments

1. Although the abstract is informative, the findings are not sufficiently summarised in it. Therefore, it should be re-written to summarise the results of the manuscript appropriately.
2. There are several typos and grammatical mistakes, mainly in the methods section, in the manuscript that require attention.

Decision letter (RSOB-19-0035.R0)

09-Apr-2019

Dear Dr Romauch,

We are writing to inform you that the Editor has reached a decision on your manuscript RSOB-19-0035 entitled "ZINC- α 2-GLYCOPROTEIN ACTS AS INHIBITOR OF AMINE OXIDASE CONTAINING COPPER 3", submitted to Open Biology.

As you will see from the reviewers' comments below, there are a number of criticisms that prevent us from accepting your manuscript at this stage. The reviewers suggest, however, that a revised version could be acceptable, if you are able to address their concerns. If you think that you can deal satisfactorily with the reviewer's suggestions, we would be pleased to consider a revised manuscript.

The revision will be re-reviewed, where possible, by the original referees. As such, please submit the revised version of your manuscript within four weeks. If you do not think you will be able to meet this date please let us know immediately.

When submitting your revised manuscript, please respond to the comments made by the referee(s) and upload a file "Response to Referees" in "Section 6 - File Upload". You can use this to document any changes you make to the original manuscript. In order to expedite the

processing of the revised manuscript, please be as specific as possible in your response to the referee(s).

Please see our detailed instructions for revision requirements
<https://royalsociety.org/journals/authors/author-guidelines/>

Sincerely,

The Open Biology Team
 mailto: openbiology@royalsociety.org

Reviewer(s)' Comments to Author(s):

Referee: 1

Comments to the Author(s)

The manuscript is dealing with the biological functions of a soluble protein known from a while to be related with cachexia, therefore with energy balance, the Zinc-alpha2-glycoprotein (ZAG). Reported data show binding of ZAG to the membrane form of Vascular adhesion protein-1 (VAP-1), an adhesion molecule endowed with a semicarbazide-sensitive amine oxidase (SSAO) activity. SSAO/VAP-1 is abundant in adipocytes, vessels, and sites of inflammation, and supposed to generate reactive oxygen species once oxidizing amines. The interaction between these two proteins leads to the inhibition of SSAO activity. Thereby, several biogenic amines being substrates of SSAO/VAP-1, have likely their half-life increased and can longer stimulate their receptors. This seems to be the case of octopamine, which activates beta3-adrenoceptors and therefore increases lipolysis in adipocytes, probably making the link between ZAG and beta3-adrenoceptor activation proposed by various previous reports. The present findings support the conclusions of the author and surprisingly suggest that two endogenous multifunctional proteins might have uncovered incidences on diabetes and obesity. However, such novel important message deserves a better presentation of the manuscript, which suffers from numerous minor defects, and which needs in-deep revision, into the opinion of reviewer, alongside three other concerns.

A) As far as obesity or diabetes are concerned, a major concern could be related about the absence of clinically relevant observations, but the message being so novel, the preclinical observations are largely sufficient for publication in open Biology.

B) Another concern is that taking into account the variety of methods used in this report, it looks strange to the reviewer that the submitted work, which is not a review, is signed by only one author?

c) Then, the English style mixing past and present, and using commas too scarcely, has to be improved substantially; as it is the case of the last sentence of introduction ... and the first of conclusion. Moreover, the manuscript has not numbered lines, which did not facilitate building the necessary list of amendments, which is given below in a non-exhaustive manner:

- 1) check spelling of concavalin A in p2, first paragraph.
- 2) is ref 44 really dealing with the interplay between ZAG and β 3-adrenoceptors?
- 3) the abbreviations GST and PNGase are not introduced in any part of the Manuscript (Ms). Moreover GST-tag is even less described

- 4) Fig 1 is unreadable, at least at the reduced size chosen for the manuscript: For instance the ZAG indicated in a yellow circle is much lower than Times/Helvetica 8. Moreover, this figure has nothing to do in the Result section, and likely will be better located in Methods.
- 5) Figs 2 and 3 also contain panels that are too much reduced in their final appearance. The reviewer needed a X300 enlargement to distinguish items in several panels.
- 6) errors in title of fig 3. Can author(s) replace(s) by: ZAG modifies AOC3 activity ?
For panels B, D and related, the units of Y-axis are not correct, they seem to be in pmol H₂O₂ /mg protein/min in reality ? More precisely, authors have to consider that activity is not normalized per mg of proteins as proposed, but per mg of AOC3 recombinant protein, since when adding ZAG to AOC3 protein, the amount of protein is increasing in the well and subsequently the proportion of active amine oxidase is decreasing, therefore leading to a lowered velocity per mg of (total) proteins, which should be observable with the addition of any other protein, even non-reacting with SSAO activity. Otherwise, it must be stated that the same amount of enzyme is distributed in control and assays in the presence of increasing doses of ZAG. Also applies for endogenous SSAO activity of 3T3 and coronary cells, for Fig 4B, and for supplemental data. Authors have to indicate in X-axes that they tested 12.5 ng ZAG and not 12.5.
- 7) Chemical name of LJP1586 has to be provided and source indicated in Methods.
- 8) Fig 4B the increase of SSAO activity by plasma fractions devoid of ZAG is even greater than the inhibition by the soluble protein. Can the authors add other tests on this effect rather than building an hypothesis on stimulatory effects of albumin to state more unambiguously on this issue? Supplemental data needs much comments, especially the repeated lacks of dose-response.
- 9) Fig 5 also suffers of too much numerous and reduced/small panels. "Noradrenaline" must be corrected for each occurrence/. As the same research experiment was performed in panels D and G, reviewer proposes to separate panels G and H as a novel figure. This will show with insistence that G and H were (perhaps?) performed in parallel and will authorize the authors to confirm their findings originally observed in 5D in a further approach that consisted in comparing a SSAO inhibitor and ZAG. Otherwise, remove these two panels and make a fusion of their common observations, since for scientific publications, a rule is not to show repeatedly the same results of a same experiment, save if normalization of data has an influence on interpretation...
- 10) Obviously, the lack of inhibition by plasma from ZAG ko mice indicates in a clearer manner what component of plasma is involved in the inhibition of AOC3. Why not testing addition of BSA to ZAG ko plasma?
- 11) Conclusion/interpretation of the data are given at the end of the figure legends, whilst this is generally indicated in Results section referring to the figure. Thus, most of the last sentences of the figure legends have to be reinserted in Results or deleted when already present.
- 12) The simple observational statement at the end of p10 needs further attention: Authors should comment why ZAG is not behaving as the soluble SSAO inhibitor regarding the improvement of lipolysis in response to low or high doses of octopamine?
- 13) Regarding the influence of ZAG glycosylation state on SSAO inhibition, the test of human plasma would have brought more clinically relevant info than the comparison of multiple forms with different glycosylation patterns from diverse sources, even including human embryo. Facing to a so complex approach of various MW forms of ZAG, the bullet sentence "The inhibitory effect clearly depends on the expression host" in p 13 is not so informative as it appears and needs further clarification. On the other hand, the first sentence of discussion is revealing an objective that was not declared before and that does not seem to be supported by provided data. Thus, the search for a "ZAG receptor" should be presented with more caution or replaced by the term "interaction partner" more adequately used in the discussion.
- 14) The passage in discussion about hydrogen peroxide-dependent signalling is too long since not any experiment, except the measurement of its production during amine oxidation, has been performed in this way.
- 15) many typo errors have to be checked throughout the Ms.
- 16) Lastly, not all the doi are provided in the reference list.

Referee: 2

Comments to the Author(s)

Zinc-alpha2-glycoprotein (ZAG) plasma protein is an important biomarker for the diagnosis of cachexia. ZAG is known to cause progressive weight loss by increasing β -adrenergic receptor-mediated lipolysis in adipocytes. In this manuscript, the authors isolated amine oxidase (AOC)3 as a ZAG interacting protein. AOC3 is a copper-containing cell surface protein, expressed mainly in adipocytes and endothelial cells, that can affect both lipolysis and extravasation of immune cells. Further, they showed that ZAG modulates AOC3-dependent lipolysis in adipocytes and leukocyte transmigration by acting as an allosteric inhibitor of AOC3. Based on these results, the authors concluded that ZAG also acts on lipolysis in a β -adrenergic signalling independent manner. The findings are interesting but addressing the following comments will strengthen the manuscript.

Comments

1. Although the abstract is informative, the findings are not sufficiently summarised in it. Therefore, it should be re-written to summarise the results of the manuscript appropriately.
2. There are several typos and grammatical mistakes, mainly in the methods section, in the manuscript that require attention.

Author's Response to Decision Letter for (RSOB-19-0035.R0)

See Appendix A.

RSOB-19-0035.R1 (Revision)

Review form: Reviewer 1 (Christian Carpené)

Recommendation

Accept with minor revision (please list in comments)

Is the length of the paper justified?

Yes

Should the paper be seen by a specialist statistical reviewer?

No

Is it clear how to make all supporting data available?

No

Is the supplementary material necessary; and if so is it adequate and clear?

No

Do you have any ethical concerns with this paper?

Yes

Comments to the Author

Most of the concerns have been carefully and adequately treated, making this (long) manuscript (MS) more suitable for publication.

The reviewer, although impressed by the quantity of work is still disappointed by the fact that the manuscript is signed only by one author, and especially respectfully disagrees with the style using 'I attempted to purify ...' or 'I tested whether ...' in lines 140, 179, 200 and elsewhere in the MS. Can at least these sentences be replaced by impersonal constructions?

Although deeply revised and improved, this version revealed small problems such as the lack of abbreviation for Sulfo SBED line 106, and reviewer suspected a mismatch with the ref 46 in line 414 (not dealing with leukocyte adhesion as indicated). Lastly, the long reference list does not exhibit homogeneous style, since still having full titles or abbreviations for journal names.

Therefore, small editorial issues should be treated before dissemination, probably at the step of proofreading.

Review form: Reviewer 2 (Christian Carpene)**Recommendation**

Accept as is

Comments to the Author

None

Decision letter (RSOB-19-0035.R1)

24-Jun-2019

Dear Mr Romauch,

We are pleased to inform you that your manuscript RSOB-19-0035.R1 entitled "ZINC- α 2-GLYCOPROTEIN ACTS AS INHIBITOR OF AMINE OXIDASE COPPER CONTAINING 3" has been accepted by the Editor for publication in Open Biology. The reviewer(s) have recommended publication, but also suggest some minor revisions to your manuscript. Therefore, we invite you to respond to the reviewer(s)' comments and revise your manuscript.

Please submit the revised version of your manuscript within 7 days. If you do not think you will be able to meet this date please let us know immediately and we can extend this deadline for you.

- 1) A text file of the manuscript (doc, txt, rtf or tex), including the references, tables (including captions) and figure captions. Please remove any tracked changes from the text before submission. PDF files are not an accepted format for the "Main Document".
- 2) A separate electronic file of each figure (tiff, EPS or print-quality PDF preferred). The format should be produced directly from original creation package, or original software format. Please note that PowerPoint files are not accepted.
- 3) Electronic supplementary material: this should be contained in a separate file from the main text and meet our ESM criteria (see <http://royalsocietypublishing.org/instructions-authors#question5>). All supplementary materials accompanying an accepted article will be treated as in their final form. They will be published alongside the paper on the journal website and posted on the online figshare repository. Files on figshare will be made available approximately one week before the accompanying article so that the supplementary material can be attributed a unique DOI.

Online supplementary material will also carry the title and description provided during submission, so please ensure these are accurate and informative. Note that the Royal Society will not edit or typeset supplementary material and it will be hosted as provided. Please ensure that the supplementary material includes the paper details (authors, title, journal name, article DOI). Your article DOI will be 10.1098/rsob.2016[last 4 digits of e.g. 10.1098/rsob.20160049].

- 4) A media summary: a short non-technical summary (up to 100 words) of the key findings/importance of your manuscript. Please try to write in simple English, avoid jargon, explain the importance of the topic, outline the main implications and describe why this topic is newsworthy.

Images

Data-Sharing

It is a condition of publication that data supporting your paper are made available. Data should be made available either in the electronic supplementary material or through an appropriate repository. Details of how to access data should be included in your paper. Please see <http://royalsocietypublishing.org/site/authors/policy.xhtml#question6> for more details.

Data accessibility section

Sincerely,
The Open Biology Team
mailto:openbiology@royalsociety.org

Reviewer(s)' Comments to Author:

Referee: 2

Comments to the Author(s)
None

Referee: 1

Comments to the Author(s)

Most of the concerns have been carefully and adequately treated, making this (long) manuscript (MS) more suitable for publication.

The reviewer, although impressed by the quantity of work is still disappointed by the fact that the manuscript is signed only by one author, and especially respectfully disagrees with the style using 'I attempted to purify ...' or 'I tested whether ...' in lines 140, 179, 200 and elsewhere in the MS. Can at least these sentences be replaced by impersonal constructions?

Although deeply revised and improved, this version revealed small problems such as the lack of abbreviation for Sulfo SBED line 106, and reviewer suspected a mismatch with the ref 46 in line 414 (not dealing with leukocyte adhesion as indicated). Lastly, the long reference list does not exhibit homogeneous style, since still having full titles or abbreviations for journal names.

Therefore, small editorial issues should be treated before dissemination, probably at the step of proofreading.

Author's Response to Decision Letter for (RSOB-19-0035.R1)

See Appendix B.

Decision letter (RSOB-19-0035.R2)

04-Jul-2019

Dear Mr Romauch

We are pleased to inform you that your manuscript entitled "ZINC- α 2-GLYCOPROTEIN IS AN

INHIBITOR OF AMINE OXIDASE COPPER CONTAINING 3" has been accepted by the Editor for publication in Open Biology.

Article processing charge

Please note that the article processing charge is immediately payable. A separate email will be sent out shortly to confirm the charge due. The preferred payment method is by credit card; however, other payment options are available.

Sincerely,

The Open Biology Team
mailto: openbiology@royalsociety.org

Appendix A

Letter for Reviewer 1:

Dear Reviewer!

Thank you for reviewing the manuscript (MS) and I have to apologize for not numbering the lines which unnecessarily complicated for you the review of the MS. Your comments were very helpful and taught me in some way to improve the quality of a scientific manuscript in general. Your questions are answered in chronological order, beginning with your comments I received on the 4th of April, followed by your comments I received on the 17th of May. I will answer your questions by citing your comments first, followed by my answer.

Answers to comments received on the 4th of April, 2019

A) As far as obesity or diabetes are concerned, a major concern could be related about the absence of clinically relevant observations, but the message being so novel, the preclinical observations are largely sufficient for publication in open Biology.

B) Another concern is that taking into account the variety of methods used in this report, it looks strange to the reviewer that the submitted work, which is not a review, is signed by only one author?

I am aware of the fact that that it looks unusual having signed this work only by myself. I need to explain that this work is a long-lasting project for which only I was responsible. Since no topic-related pre-work existed on which I could build on, this project was accompanied by several technical obstacles I had to overcome. Additionally, I was involved in other projects which in the end took me a long time to finish this manuscript. This scientific paper covers the main findings of my Diploma thesis (“Isolierung eines putativen Rezeptors für Zink- α 2-Glykoprotein”, 2011), my PhD-thesis (“Deciphering Zinc-alpha 2-glycoprotein-signaling”, 2015) and the first year of my engagement as a Post-Doc. My work on ZAG began with a cancer-related grant (Hans und Blanca Moser Stiftung) I received after earning my first doctoral degree in medicine. In parallel I studied chemistry and working on my cancer project made me decide to dedicate myself on finding a novel interaction partner of ZAG. This effort resulted in the identification of AOC3, which is described in my Diploma thesis. This new protein-interaction was deepened during my work as PhD student, for which I earned my second doctoral degree, and the first year as Post-Doc. During this time, I perfected cross-linking experiments, optimized MS/MS protein sequencing protocol with the help of the core facility of mass-spectrometry based proteomics, UKE (Hamburg), established several new

methods for recombinant protein expression, such as the lentiviral expression system, *Komagatella pastoris*, insect cells and the Expi293F cells, designed a plethora of expression plasmids, introduced new assays, developed new expression and purification protocols, and designed cell culture experiments. There was vivid exchange with my working colleagues Gerhard Hofer (cell culture work) and Roland Viertlmayr (ion-exchange chromatography), but all presented data and described methods refer exclusively to my effort and study design. Furthermore, the whole MS was written and all intellectual input was provided by myself. According to the authorship criteria of Open Biology:

- substantial contributions to conception and design, or acquisition of data, or analysis and interpretation of data;
 - drafting the article or revising it critically for important intellectual content;
 - final approval of the version to be published; and
 - agreement to be accountable for all aspects of the work in ensuring that questions related to the accuracy or integrity of any part of the work are appropriately investigated and resolved,
- none of my working colleagues meet all four criteria.

c) Then, the English style mixing past and present, and using commas too scarcely, has to be improved substantially; as it is the case of the last sentence of introduction ... and the first of conclusion. Moreover, the MS has not numbered lines, which did not facilitate building the necessary list of amendments, which is given below in a non-exhaustive manner.

The new MS is completely new edited with the help of “Cambridge academic manuscripts” (<http://www.cambridgeacademicmanuscripts.com/>). Figure legends and Figures are found at the end of the new MS and the supplementary material

1) check spelling of concavalin A in p2, first paragraph.

The term concaavalin was removed. The sentence of the old MS:” In the seminal fluid, ZAG is localized to the surface of spermatozoa, where it is supposed to be involved in the sperm motility and capacitation process by interacting with concaavalin A”, was removed and is replaced with **(line 49-51):**

“In seminal fluid, ZAG is found on the surface of spermatozoa, where it is thought to be involved in sperm motility and capacitation.”

2) is ref 44 really dealing with the interplay between ZAG and β_3 -adrenoceptors?

This paper indirectly analyzes the β_3 -adrenergic signaling by adding propranolol, a non-selective inhibitor of β -adrenergic receptors. In an *in vitro* experiment α -SMA expression was induced in NRK-49F fibroblasts by adding TGF- β . Co-incubation with ZAG prevented α -SMA expression. ZAG is described to act by binding to the β_3 -adrenergic receptor, but addition of propranolol did not reverse ZAG's inhibited expression of α -SMA.

Therefore, the sentence:” However, inhibition of the β_3 -adrenergic receptor by propranolol, an unspecific inhibitor of β -adrenergic receptors, did not lead to any improvement, suggesting that ZAG might interact with another receptor”

is replaced with **(line 79-85):**” *In vitro* experiments revealed that TGF- β -induced expression of α -SMA can be blocked by addition of ZAG. Co-immunoprecipitation experiments showed that ZAG neither interacts with TGF- β nor its receptor, however. Furthermore, blocking ZAG signaling, which is supposedly mediated through the β_3 -adrenergic receptor, by propranolol, a non-selective antagonist of β -adrenergic receptors, did not restore TGF- β -induced α -SMA expression. This suggests that ZAG mediates its anti-inflammatory effect through a β_3 -adrenergic-independent signaling pathway.”

3) the abbreviations GST and PNGase are not introduced in any part of the MS (Ms). Moreover GST-tag is even less described

Full name of GST is introduced in **line 103₂**: GST (glutathione-S-transferase)-tag

and full name of PNGase is introduced in **line 289**: PNGase F (peptide: N-glycosidase F).

GST-tag is described in line **638-639**” [...] while the GST-tag, equipped with a cleavage site (recognised by PreScission Protease) [...].”

4) Fig 1 is unreadable, at least at the reduced size chosen for the manuscript: For instance the ZAG indicated in a yellow circle is much lower than Times/Helvetica 8. Moreover, this figure has nothing to do in the Result section, and likely will be better located in Methods.

Fig. 1 is rearranged, font sizes enlarged and inserted in “Figures”, p. 68.

Figure 1 Comparison of old and new figures. Re-arranged Figure 1 of the old MS

5) Figs 2 and 3 also contain panels that are too much reduced in their final appearance. The reviewer needed a X300 enlargement to distinguish items in several panels.

Fig. 2 and 3 of the old MS are now Fig. 1 and 2 in the new MS. Panels of Fig. 1 (p. 61) are rearranged and font sizes enlarged. Fig. 2 is split up onto Fig. 2 (p. 62) and Fig. 3 (p. 63). Panels are rearranged and font sizes enlarged.

old MS

new MS

Figure 2 Comparison of old and new figures. Re-arranged Fig.2 of the old MS, which is now Fig. 1 of the new MS

old MS

new MS

Figure 3 Comparison of old and new figures. Re-arranged Fig. 3 of the old MS, which is now Fig.2 and (+) Fig.3 of the new MS

6) errors in title of fig 3. Can author(s) replace(s) by: **ZAG modifies AOC3 activity?**

Title is replaced with “ZAG modifies AOC3 activity” in following figures:

Fig. 2, panel C

Fig. 3, panel A and B

Fig. 4, panel B and D

Fig. 8, panel C

For panels B, D and related, the units of Y-axis are not correct, they seem to be in pmol H₂O₂ /mg protein/min in reality? More precisely, authors have to consider that activity is not normalized per mg of proteins as proposed, but per mg of AOC3 recombinant protein, since when adding ZAG to AOC3 protein, the amount of protein is increasing in the well and subsequently the proportion of active amine oxidase is decreasing, therefore leading to a lowered velocity per mg of (total) proteins, which should be observable with the addition of any other protein, even non-reacting with SSAO activity.

It is correct that activity refers to mg AOC3 (50 ng) used in assays. The Y-axis title “pmol H₂O₂ /(mg protein*min)” is replaced with “pmol H₂O₂ /(mg AOC3*min)” in the following figures:

- Fig. 2, panel B, C and D

The Y-axis title “pmol benzaldehyde/(mg protein*h)” is replaced with “pmol benzaldehyde /(mg AOC3*h)” in the following figures:

- Fig. 4, panel B and D
- Fig. 8, panel C
- Supplemental Fig. 1, panel D and p. 2, Supplemental Fig. 2, panel A, B and C

Otherwise, it must be stated that the same amount of enzyme is distributed in control and assays in the presence of increasing doses of ZAG.

Following legends are extended with:” AOC3 (50 ng)” to include how much recombinant AOC3 was used:

- legend of Fig. 2, panel B, C and D
legend of Fig. 4, panel D
- Supplemental Fig. 1, panel D and p. 2, Supplemental Fig. 2, panel A and B

Also applies for endogenous SSAO activity of 3T3 and coronary cells

Y-axis title “pmol benzaldehyde/(mg protein*h)” is replaced with “pmol benzaldehyde/(mg cellular protein*h)” in following figures:

- Fig. 3, panel A and B

,for Fig 4B , and for supplemental data.

Y-title “pmol benzaldehyde/(mg protein*h)” is replaced with “pmol benzaldehyde/(mg AOC3*h)”:

- Fig. 4, panel B and D
- Supplemental Fig. 1, panel D and p. 2, Supplemental Fig. 2, panel A, B and C

Authors have to indicate in X-axes that they tested 12.5 ng ZAG and not 12,5.

Comma as decimal separator is replaced with a point in following figures:

- Fig. 2, panel C
- Fig. 3, panel A and B

7) Chemical name of LJP1586 has to be provided and source indicated in Methods.

- line 159-160, chemical name of LJP1586: (Z-3- fluoro-2-(4-methoxybenzyl)-allylamine hydrochloride)
- line 684, source of chemical compound: La Jolla Pharmaceuticals

8) Fig 4B the increase of SSAO activity by plasma fractions devoid of ZAG is even greater than the inhibition by the soluble protein. Can the authors add other tests on this effect rather than building an hypothesis on stimulatory effects of albumin to state more unambiguously on this issue?

Supplemental Fig. 1 and Supplemental fig. 2 experimentally explain in detail what might contribute to increasing AOC3 activity by addition of no-ZAG IEX fractions. In the **new MS** enhancement of recombinant AOC3 is explained by the following paragraph (**lines 207-231**):

“The ZAG-positive fractions reduced recombinant AOC3 in a dose-dependent manner. However, the control IEX fractions, which contained no ZAG, enhanced recombinant AOC3 activity in a dose-dependent manner, rather than having the expected neutral effect. This stimulatory effect is probably due to both endogenous AOC3 activity and plasma components. First, murine plasma (except that of AOC3 k.o. mice) contains endogenous amine oxidase activity, which can be blocked by the inhibitor LJP1586 (Supplemental Fig. 1, A, B and C). Plasma-derived AOC3 activity results from release of the membrane-bound enzyme from cells by metalloprotease activity [51–53]. However, measurement of amine oxidase activity of IEX fractions – either containing or not containing ZAG – before adding recombinant AOC3 revealed no endogenous activity (Supplemental Fig. 2, C). Second, incubation of recombinant AOC3 with plasma of wt, AOC3 k.o. and ZAG k.o. mice enhanced AOC3 activity ~3-fold (Supplemental Fig. 1, D). Therefore, a plasma component found in all three genotypes must be responsible for enhancing AOC3 activity. Third, the IEX fractions lacking ZAG (D3 and D4) correspond to the major protein peak of the IEX profile, which mostly derives from albumin. Incubating recombinant AOC3 with fatty acid-free bovine serum albumin (BSA) also enhanced recombinant AOC3 activity to the same extent as murine plasma (Supplemental Fig. 2, A). Fourth, the literature describes a low molecular-weight plasma component (3.8 kDa), which in

combination with lysophosphatidylcholine (LPC) boosts AOC3 activity [54]. LPC makes up to 4-20% of total plasma phospholipid content [55] and albumin is an important LPC storage protein [56]. Therefore, the IEX fractions lacking ZAG may contain the AOC3-activating plasma component, which is fully active in the presence of LPC. Notably, incubation of human lung-microsomal AOC3 with filtered and lyophilized human plasma (FLHP) enhances AOC3 activity up to 5-fold [54], which is similar to the effect of adding 200 μ l of the IEX fractions lacking ZAG to recombinant AOC3 (**Error! Reference source not found., B).**)”

Supplemental data needs much comments, especially the repeated lacks of dose-response.

Content of Supplemental Figures 1 and 2 of the old MS are summarized, extended and found as Supplemental Figure 1 and 2.

New Supplemental Figure 3 deals with a further experiment to answer the question **#12: Authors should coment why ZAG is not behaving as the soluble SSAO inhibitor regarding the improvement of lipolysis in response to low or high doses of octopamine?**

All Figures are commented in detail.

9) Fig 5 also suffers of too much numerous and reduced/small panels. "Noradreanline" must be corrected for each occurrence/. As the same research experiment was performed in panels D and G, reviewer proposes to separate panels G and H as a novel figure. This will show with insistence that G and H were (perhaps?) performed in parallel and will authorize the authors to confirm their findings originally observed in 5D in a further approach that consisted in comparing a SSAO inhibitor and ZAG. Otherwise, remove these two panels and make a fusion of their common observations, since for scientific publications, a rule is not to show repeatedly the same results of a same experiment, save if normalization of data has an influence on interpretation...

Fig. 5 is rearranged and font sizes are enlarged. Every occurrence of “Noradreanline” is replaced with “Noradrenaline”. Fig. 5 in the old MS is split into **Fig. 5, p. 64** and **Fig. 6, p. 65** in the new MS. You are right in assuming that cell experiments were performed in parallel. For

not repeating the same result, Fig. 6, p.65 is introduced to show the direct comparison between LJP1586 and ZAG.

Figure 4 Comparison of old and new figures. Fig. 5 of the new MS is newly arranged, font sizes are enlarged and noradrenaline replaced with noradrenaline. Panel G and H of Fig. 5 of the old MS are taken out now building Fig. 6 of the new MS.

10) Obviously, the lack of inhibition by plasma from ZAG ko mice indicates in a clearer manner what component of plasma as involved in the inhibition of AOC3. Why not testing addition of BSA to ZAG ko plasma?

I guess that the experimental suggestion:” Why not testing addition of BSA to ZAG ko plasma”, refers to Fig. 4, D. I was thinking hard how this experiment could fit to the finding that ZAG k.o. plasma shows no inhibitory effect on recombinant AOC3. The difference in recombinant AOC3 activity, incubated with plasma IEX-fractions of wt and ZAG k.o. mice, only relates to IEX-fractions C12 and D1. Subsequent IEX fractions of both wt and ZAG k.o. plasma show no different effect on recombinant AOC3 activity. To prevent any misunderstanding, panel D of Fig. 4 (p. 63) now only shows IEX fractions C12 and D1, comparing the inhibitory effect on recombinant AOC3 (Figure 5).

Figure 5 Comparison of panel B and D of Fig. 4. of the old MS (manuscript) and the new MS. Inhibition of recombinant AOC3 by ZAG-IEX fraction and no ZAG-IEX fraction was extended by lower concentrations. To emphasize the difference between wt and ZAG k.o. only IEX fractions of C12 and D1 are represented in the new panel D, Fig. 4. of the new MS

11) Conclusion/interpretation of the data are given at the end of the figure legends, whilst this is generally indicated in Results section referring to the figure. Thus, most of the last

sentences of the figure legends have to be reinserted in Results or deleted when already present.

Following sentences of figure legends were removed and introduced in the Results section:

- **Old MS**, p. 4, Figure legend 2: “Using Streptavidin antibody labelled GST-tag (*) and human and murine GST-ZAG (***) can be detected. The band above 75kDa (***) represents the unknown protein, which got a biotin-tag transferred after reducing the disulfide bridge of Sulfo-bred. (...) Only the GST-tag and murine and human GST-ZAG can be detected. (...) . GST-AOC3 bound plasma derived ZAG. GST alone did not.”

Comments of Figure legend 1 (**new MS**) are introduced or already present in Results:

lines 111-119:” Using streptavidin, one band was detected using GST-tag (**Error! Reference source not found.**, Aa, lane 1) as bait protein and three bands were detected using GST-mZAG or GST-hZAG as bait proteins (**Error! Reference source not found.**, Aa, lane 2 and 3). The lowest band at ~26 kDa (kildodalton) represents the labeled GST-tag (*) (**Error! Reference source not found.**, Aa, lanes 1-3) and was found in the control and samples incubated with GST-ZAG (**). This is due to loss of the GST-tag, which could not be completely prevented during overexpression of GST-ZAG in *E. coli*. The band at ~66kDa represents labeled GST-ZAG (***) (**Error! Reference source not found.**, Aa, lanes 2 and 3). The band at ~80 kDa represents a hitherto-unknown protein X (***), to which a biotin tag was transferred after reducing the crosslinker molecule with β -mercaptoethanol (**Error! Reference source not found.**, Aa, lanes 2 and 3)..”

lines 123-125:” After stripping, the WB membrane was reprobed with α -GST antibody, when only GST-tag (**Error! Reference source not found.**, Ab, lane 1, 2 and 3), GST-tagged murine ZAG (**Error! Reference source not found.**, Ab, lane 2) and GST-tagged human ZAG (**Error! Reference source not found.**, Ab, lane 3) were detected.”

lines 146-147:” A WB of the eluate fraction revealed that GST-AOC3 bound ZAG from murine plasma, whereas GST alone did not.”

- **Old MS**, p. 6, Figure legend 3:” Recombinant or endogenous AOC3 is incubated with synthetic substrate benzylamine or [14C]-benzylamine. For non-radioactive measurement activity is measured by H₂O₂ which oxidizes Amplex Red to its fluorescent analogue resorufin in the

presence of horse radish peroxidase (HRPO). Using [¹⁴C]-benzylamine as substrate activity corresponds to generated amount of [¹⁴C]-benzaldehyde. Per molecule benzylamine one molecule H₂O₂ and one molecule NH₃ are generated. LJP1586 serves as inhibitor. (...) At different concentrations highest activity is reached at 100 μM benzylamine. (...) Strongest inhibition is observed at equimolar ratio. (...) Increasing amounts of ZAG lead to reduction of V_{max}, whereas K_m remains constant. (...) ZAG acts on both human and murine AOC3 and inhibits as strong as LJP1586.”

Comments of Figure legend 2 (**new MS**) are introduced or already present in Results:

lines 153- 161:” For activity measurements, recombinant or endogenous AOC3 is incubated with the synthetic substrate benzylamine or [¹⁴C]-benzylamine. Non-radioactive assays measure H₂O₂, which oxidizes Amplex Red to its fluorescent analog resorufin in the presence of horse radish peroxidase (HRPO). Using [¹⁴C]-benzylamine as substrate, the activity corresponds to the amount of [¹⁴C]-benzaldehyde generated. For each molecule of benzylamine, one molecule of H₂O₂ and one molecule of NH₃ are generated. LJP1586 (Z-3-fluoro-2-(4-methoxybenzyl)-allylamine hydrochloride) serves as an inhibitor. To investigate whether ZAG can modulate AOC3 activity, both GST-tagged AOC3 and ZAG of murine origin were purified from lentivirally transduced HEK293 cells and the GST-tag was removed.”

lines 163-165:” In a first attempt, activity assays were performed using Amplex Red reagent (**Error! Reference source not found.**, A). A saturation curve using benzylamine as substrate revealed the highest activity at 100 μM (**Error! Reference source not found.**, B).”

lines 165-168:” To characterize the interaction between AOC3 and ZAG, both proteins were mixed at different molar ratios. A stepwise increase in the concentration of recombinant ZAG led to a stepwise decrease in recombinant AOC3 activity. The strongest inhibition was observed at a molar AOC3/ZAG ratio of ~ 1:1 (25 ng ZAG) (**Error! Reference source not found.**, C).”

lines 169-172: Next, the mechanism of inhibition was investigated by generating a Michaelis-Menten plot, which revealed that V_{max} (maximum velocity) decreases, whereas K_m (i.e. the Michaelis-Menten constant: substrate concentration at half-maximum velocity) remains almost constant, with rising ZAG concentrations (**Error! Reference source not found.**, D).

lines 183-184: The activity of 3T3-L1-derived AOC3 was effectively reduced as the amount of recombinant ZAG was increased.

p. 9, lines 190-193: Furthermore, the inhibitory effect of recombinant ZAG on HCAEC AOC3 (**Error! Reference source not found.**, B) confirmed the similarity between murine and human AOC3, underlining the crosslinking results obtained with SGBS cell membranes and indicating that the ZAG-AOC3 interaction also plays an important role in humans.

- **Old MS**, p. 7, Figure legend 4:” Fractions C10, C12 and D1 show a signal between 37 and 50 kDa. (...) Strongest inhibition is already observed with 50 µl. Following fractions (D2, D3) increase AOC3 activity. (...) WB of wt IEX-fractions show a ZAG signal between 37 and 50 kDa, ZAG k.o. IEX-fractions do not. (...). Inhibition of recombinant AOC3 activity can be observed for ZAG containing fractions (C12, D1) of wt plasma. IEX fractions (C12, D1) of ZAG k.o. plasma does not inhibit recombinant AOC3 activity.“

Comments of Figure legend 4 (new MS) are introduced or already present in Results:

lines 203-206: Fractions C12 and D1 showed a signal between 37 kDa and 50 kDa, which corresponds to murine plasma ZAG (**Error! Reference source not found.**, A). ZAG-containing fractions (C12 and D1) were pooled, as were fractions without any ZAG (D3 and D4) as controls, and incubated with 50 ng recombinant AOC3 (**Error! Reference source not found.**, B)

lines 240-242:” ZAG-containing fractions (C12 and D1) of wt plasma were identified by WB using α -ZAG antibody. WB of corresponding fractions of ZAG k.o. plasma show no signal.”

lines 235-237:” As before, ZAG-containing fractions of wt plasma reduced benzylamine catalysis by AOC3, whereas the corresponding fractions of ZAG k.o. plasma did not (**Error! Reference source not found.**, D).

- **Old MS**, p. 9, Figure legend 5:” Significant difference compared to control is observed after 18 hours. (...) Strongest lipolytic effect can be detected for noradrenaline and octopamine. (...) Activity can be detected against tyramine, histamine, dopamine, cadaverine, cysteamine, ethanolamine, octopamine, putrescin, spermidine, isopentylamine and benzylamine. (...). Only octopamine stimulated lipolysis can be enhanced by addition of LJP1586 (10 µM).”

Comments of Figure legend 5 and 6 (new MS) are introduced or already present in Results:

lines 244-250:” Compared to ZAG, isoproterenol significantly enhanced glycerol release already within the first thirteen minutes. ZAG, GST and LJP1586 showed no such effect.

Incubating differentiated 3T3-L1 cells with ZAG (50 µg/ml), GST (50 µg/ml) and LJP1586 (10 µM) for several hours revealed that, although ZAG showed a lipolytic effect, this did not occur until twelve hours (**Error! Reference source not found.**, B). Therefore, ZAG definitely does not behave as a classical β-adrenergic agonist such as isoproterenol and another mechanism must be involved in ZAG-stimulated lipolysis, most likely involving AOC3.”

lines 263-264:” Notably, noradrenaline and octopamine both strongly stimulated lipolysis, but only octopamine was converted by AOC3.”

lines 258-260:” The strongest activity was observed with tyramine, histamine, dopamine, cadaverine, cysteamine, ethanolamine, octopamine, putrescin, spermidine, isopentylamine and benzylamine (**Error! Reference source not found.**, C).”

lines 272-276:” In the case of noradrenaline and isoproterenol, the addition of LJP1586 did not enhance glycerol release (**Error! Reference source not found.**, E and F). This is in line with the observations that noradrenaline is not converted by AOC3 and isoproterenol contains no primary amine. However, the presence of LJP1586 (10 µM) or ZAG (50 µg/ml) boosted octopamine-stimulated lipolysis (Fig. 6, A and B).”

- **Old MS**, p. 11, Figure legend 6:” Addition of the o-glycosylation inhibitor leads to no difference. (...) Wt ZAG, and other glycomutants, overexpressed in different expressions hosts show a different inhibitory effect at the same concentration (50 µg/ml).”

Comments of Figure legend 7 and 8 (**new MS**) are introduced or already present in Results:

lines 307-311:” Since O-glycosylation might also affect the size of the protein, another overexpression experiment was performed with GST-ZAG in the presence of the inhibitor benzyl-2-acetamido-2-deoxy-α-D-galactopyranoside (BAGN). No effect on the size of the protein was observed, however, which underlines the notion that size differences depend on N-glycosylation events (**Error! Reference source not found.**, B)..”

lines 353-358:” After GST affinity purification, PreScission Protease digestion and dialysis, all proteins were adjusted to a concentration of 50 µg/ml and incubated with HEK293 stably expressing AOC3 with a transmembrane domain, i.e. located on the surface of the cell (**Error! Reference source not found.**, B2). Comparing the wt forms showed that HEK293 cell-derived

ZAG inhibited AOC3 activity, whereas the Expi293F cell-derived ZAG did not (**Error! Reference source not found.**, C)..”

12) The simple observational statement at the end of p10 needs further attention: Authors should comment why ZAG is not behaving as the soluble SSAO inhibitor regarding the improvement of lipolysis in response to low or high doses of octopamine?

This behavior is discussed by a new paragraph in Discussion section and supported by another experiment added as Supplementary Fig. 2.

lines 486-515:” Regarding octopamine-stimulated lipolysis in the presence of LJP1586 and ZAG (**Error! Reference source not found.**), it is notable that, compared with LJP1586, ZAG loses its stimulatory effect at higher octopamine concentrations. This difference in behavior of ZAG and LJP1586 might reflect different types of inhibition. LJP1586 is a small molecule inhibitor that enters the catalytic site of the enzyme and is highly selective for AOC3 [30,106,107]. ZAG behaves like an allosteric inhibitor, i.e. it binds away from the active site, and reduces substrate affinity. On the one hand, AOC3-derived H₂O₂ is described as affecting its own enzyme activity [108,109]: the crystal structure of human AOC3 reveals a vicinal disulfide bridge [49], which is suggested to serve as a redox switch, possibly inducing a conformational change [110]. On the other hand, human ZAG contains one disulfide bridge in its MHC-fold and one inter-sheet disulfide bridge in the immunoglobulin (Ig)-like domain. Oxidation of disulfides by H₂O₂ and one- or two-electron oxidants at physiological pH results in the formation of disulfide monoxides or disulfide dioxides, which further leads to cleavage of disulfides and the formation of sulfonic acid [111–113]. Notably, copper-containing amine oxidases also form hydroxyl radicals (one-electron oxidants) due to the reaction between H₂O₂ and reduced copper [114]. Extracellular proteins mainly contain disulfides [115] and are exposed to a higher level of ROS in general [116]. Modification of the disulfides in receptors and plasma proteins is involved in protein stability [117], protein oligomerization [118], the transformation of biological function [119,120] and receptor-ligand interaction [121]. A similar interplay between ROS signaling, ligand recognition and protein-protein interaction is imaginable for AOC3 and ZAG, which could restrict the inhibitory function of ZAG. To test whether AOC3 activity affects protein-protein interaction, wt and different glycoforms of recombinant ZAG were incubated with benzylamine in the presence or absence of recombinant

AOC3 (Supplemental Fig. 3). In the presence of AOC3, only wt ZAG shifts to a higher molecular weight, whereas in the absence of AOC3 it does not. By contrast, incubation with H₂O₂ induces a shift in wt ZAG, irrespective of whether AOC3 is present. This could hint at an oxidation-dependent oligomerization of ZAG, influencing AOC3/ZAG and/or ZAG/ZAG protein-protein interaction, in which glycosylation plays an additional role. ZAG oligomerization could serve as a self-regulatory mechanism, and explain why a complete inhibition of activity was never observed at an equimolar ratio of both proteins (**Error! Reference source not found.**, C), as well as why ZAG loses its lipolysis-stimulatory effect (**Error! Reference source not found.**).”

.13) Regarding the influence of ZAG glycosylation state on SSAO inhibition, the test of human plasma would have bring more clinical relevant info than the comparison of multiple forms with different glycosylation pattern from diverse sources, even including human embryo.

Facing to a so complex approach of various MW forms of ZAG, the bullet sentence " The inhibitory effect clearly depends on the expression host " in p 13 is not so informative as it appears and need further clarification.

To be more precise in the formulation of the analytic approach and the lack of support of analysis of human samples, following refinements are introduced:

- The title of chapter 2.4 of the old MS:” The inhibitory potential of ZAG depends on glycosylation” is replaced with the title (**line 282**):

“The inhibitory potential of recombinant ZAG depends on glycosylation.”

- The sentence of the old MS on p. 13:” The inhibitory effect clearly depends on the expression host”, is replaced with the sentence (**line 360**):

“Hence, the inhibitory effect of recombinant ZAG depends on which expression host is used.”

This sentence is dragged some lines below. Following new paragraph explains the impact of the expression host:

lines 352-360:” To investigate how glycosylation affects the inhibition of AOC3 by ZAG, all glycomutants were overexpressed in HEK293 and Expi293F cells in parallel. After GST

affinity purification, PreScission Protease digestion and dialysis, all proteins were adjusted to a concentration of 50 µg/ml and incubated with HEK293 stably expressing AOC3 with a transmembrane domain, i.e. located on the surface of the cell (**Error! Reference source not found.**, B2). Comparing the wt forms showed that HEK293 cell-derived ZAG inhibited AOC3 activity, whereas the Expi293F cell-derived ZAG did not (**Error! Reference source not found.**, C). Although the less-glycosylated form of Expi293F cell-derived ZAG is very likely to be present with the hyperglycosylated form, its inhibitory potential is strongly reduced. Hence, the inhibitory effect of recombinant ZAG depends on which expression host is used”

On the other hand, the first sentence of discussion is revealing an objective that was not declared before and that does not seem to be supported by provided data. Thus, the search for a "ZAG receptor" should be presented with more caution or replaced by the term "interaction partner" more adequately used in the discussion.

The first sentence of discussion of the old MS:” This work aims at the identification of the ZAG receptor”, is replaced with the new sentence (**lines 370-372**):

“This work aimed to identify a new interaction partner of ZAG, which might help to explain its biological functions. Although the scientific literature is divided on the issue, most authors claim that ZAG acts via the β-adrenergic system.”

14) The passage in discussion about hydrogen peroxide-dependent signalling is too long since not any experiment, except the measurement of its production during amine oxidation, has been performed in this way.

Following passage of the old MS (p. 16-17) was removed:” For example, hormone-receptor dependent src tyrosine kinase activation stimulates NADPH oxidase and inactivates H₂O₂ decomposition by peroxiredoxin. Since only peroxiredoxins close to the membranes are inactivated by phosphorylation, signaling proteins closely related to the plasma membranes, such as protein-tyrosine phosphatases (PTP) and tyrosine kinase receptors, are oxidized by H₂O₂, whereas H₂O₂ crossing the compartmentalization is inactivated by cytosolic non-

phosphorylated (active) peroxiredoxins [26]. This mechanism possibly also accounts for AOC3 since H₂O₂ activates src kinase and inactivates PTP-1B, a negative regulator of insulin signal transduction [27,28].”

Following passage was left, to summarize which components of the lipolytic cascade could be affected (**lines 483-494**):

” Although H₂O₂ is found throughout the cell, its signaling function is restricted and transduced by compartmentalization of antioxidant enzymes such as the peroxiredoxins [98]. Accordingly, AOC3-derived H₂O₂ could interfere with enzymes involved in stimulating lipolysis. Important components of this pathway are membrane-bound adenylate cyclase (AC), which generates cyclic adenosine monophosphate (cAMP), and the catalytic subunit of protein kinase A (PKA-C). Enhanced levels of cAMP bind to the regulatory unit of PKA, thereby releasing PKA-C, which in turn phosphorylates downstream elements, inducing lipolysis [61]. H₂O₂ increases levels of G(α)i – which reduces AC activity [99] – whereas PKA-C itself is inactivated by H₂O₂ [100,101]. Hence, binding of ZAG to AOC3 on adipocytes could potentially trigger lipolysis by reducing insulinogenic concentrations of H₂O₂ or by deamination of lipolytic biogenic amines, as observed in 3T3-L1 cells incubated with recombinant ZAG (**Error! Reference source not found., A**).”

15) many typo errors have to be checked throughout the Ms.

The new MS is completely new edited with the help of “Cambridge academic manuscripts” (<http://www.cambridgeacademicmanuscripts.com/>). Figure legends and Figures are found at the end of the new MS and the supplementary material

16) Lastly, not all the doi are provided in the reference list.

For adding references, I use Citavi software and the Citation style “Open Biology (Royal society)”. I contacted the Citavi support to solve the problem, that not all listed references contain a DOI address. Since the program only lists DOI which are available through online databases, it happens that some references do not list a DOI due to lack of DOI in databases used for online research. There are papers that do not have a DOI at all, such as reference #19 has (according to PubMed). In my reference list there are 21 out of 140 papers which do not list a DOI. With respect to instructions of Open Biology author guidelines (<https://royalsociety.org/journals/authors/author-guidelines/>) concerning References: “**Please**

note that references to datasets must also be included in the reference list with DOIs where available", I hope that my reference list fulfills the criteria since DOIs are listed where available.

Additional self-suggested changes:

Fig. 6 of the old MS is split into Fig. 7 (p. 14) and Fig. 8 (p. 15) of the new MS. Panels are re-arranged and font sizes enlarged.

Figure 6 Comparison of old and new figures: Fig. 6 of the old MS is now Fig. 7 and Fig. 8 of the new MS

Answers to comments received on the 17th of May, 2019

- The supplemental Fig 1 simply confirms the presence of a soluble AOC3/VAP-1/SSAO activity in mouse plasma. Such a capacity of plasma to oxidize benzylamine has been

repeatedly reported in the literature, of which it is just worth mentioning the work of F. Boomsma and colleagues that evidenced in humans the variations of plasma SSAO in diverse pathologies and situations:

You are right. I cited literature of Boomsma et al. in the new supplemental material (SM).

- Since the author claims that ZAG inhibits AOC3 activity, it would be of much greater interest to compare plasma SSAO activity in wt and ZAG-ko mice, before presenting the activities found in different fractions (Fig 4D). The lack of ZAG is expected to improve/unmask the soluble AOC3-dependent activity found in plasma, therefore to increase the overall capacity of plasma oxidation of bza:

In the new SM I added the direct comparison of plasma of wt, AOC3 k.o. and ZAG k.o. plasma (Supplemental Fig. 1, A, B and C). Lack of ZAG does not lead to a significant rise of plasma derived AOC3 activity compared to wt plasma.

- The supplemental Fig 2 does not show any dose-dependent effect for BSA and for murine plasma on benzylamine oxidation. That explains why the term "repeated lacks of dose response" was used. The study of a larger range of doses is highly recommended here (there is only a 50X factor between the lowest and the higher and the bza oxidation remains the same!). Whether the author extends the investigation and add novel doses to this fig S2, it must be noted that it is more conventional to represent dose-responses with the lower dose on the left of the graph (as in Fig 4B):

Now I clearly understand your idea. I tested the effect of plasma (of all three genotypes wt, AOC3 k.o. and ZAG k.o.) and BSA on recombinant AOC3 extensively and I took the very essence of experiments which looked important for me to explain the higher activity of recombinant AOC3, referring to panel B of Fig. 4. I intended to show what is the lowest concentration needed for maximal activation and that even higher concentrations do not further enhance AOC3 activity. According to your suggestions the diagram demonstrating the effect of BSA on recombinant AOC3 is changed and integrated in the new SM (Supplementary Fig. 2, A):

- **Moreover, the experiment lacks of negative controls consisting in determining bza oxidation with the increased doses of these factors only, without any recombinant AOC3 added. Indeed, this will allow discarding / demonstrating that the presence of the copper-containing amine oxidase alters bza oxidation induced by other plasma proteins:**

The negative control of Fig. 4, panel B refers to the blank (GST instead AOC3) which was subtracted of the experimental data received with AOC3, to be sure that the activity only refers to recombinant AOC3 activity and might not relate to any unspecific activity with increasing doses. I added the negative control (blank activity) in Supplementary Fig. 2, C:

- On the other hand, in the figure 4B, an intermediate inhibition with a smaller amount of "ZAG-like activity" than that found in 50 µL eluate is also welcome since in this situation there is still not evidence of a dose dependency: inhibition of bza oxidation is max from 50 to 200 µL. By contrast, the eluted fractions without ZAG were dose-dependently activating/stabilizing AOC3 activity. This demonstrate at least that the range for dose-dependency is different for ZAG (supposed inhibition) and for BSA, AOC3 or other plasma amine oxidases (leading to bza oxidation). A dose dependency is well demonstrated for the latter only, while the message of the present paper focuses interest on the former:

Indeed, lower concentrations of both ZAG-IEX fractions and no ZAG-IEX fractions were also tested and are now integrated in the new diagram of Fig. 4, panel B.

- In fact, the concern of Figs 4B & 4D is that they are not informative enough about the volume of plasma necessary to collect 50µL of ZAG-containing Ion Exchange fraction in order to inhibit AOC3, since the reader has no idea of the yield of the separation /elution process.

Moreover in Fig 4D ,one of the fractions corresponding to the strongest inhibition of amine oxidase activity suspected to be ZAG-enriched in wt only (D1), is also the one that gives the lowest activity in ZAG-ko plasma (when compared with D5 at least). This requires a comment about other "ZAG-like" inhibitory actors that may circulate in blood, or more likely about the distribution among fractions of factor(s) that mimic or stabilize recombinant AOC3 activity (albumin, LOLXL)? By adding BSA to a plasma that lacks ZAG , the BSA improvement of bza oxidation is expected to be greater in ZAGko than in wt (as told in point 1).

The amount of plasma needed for performing this experiment is now mentioned in Material and Methods section of the new MS (lines 765-767):” Protein from 2 ml collected murine plasma was desalted and rebuffered in 10 mM Tris-HCl, pH 8 using a PD-10 desalting column (GE Healthcare) and further diluted to a final volume of 20 ml using 10 mM Tris-HCl, pH 8.”

Concerning Fig. 4 panel D of the old MS, I fully agree that the slight rise of activity along successive fractions rather relies on non-defined boosting plasma components, albumin and increasing salt concentration, of the latter fractions than on inhibitory components of previous fractions. I discuss this phenomenon in the Results section especially focusing on this

observation related to panel B, combined with additional diagrams in the supplementary material (**lines 206-230**):

“The ZAG-positive fractions reduced recombinant AOC3 in a dose-dependent manner. However, the control IEX fractions, which contained no ZAG, enhanced recombinant AOC3 activity in a dose-dependent manner, rather than having the expected neutral effect. This stimulatory effect is probably due to both endogenous AOC3 activity and plasma components. First, murine plasma (except that of AOC3 k.o. mice) contains endogenous amine oxidase activity, which can be blocked by the inhibitor LJP1586 (Supplemental Fig. 1, A, B and C). Plasma-derived AOC3 activity results from release of the membrane-bound enzyme from cells by metalloprotease activity [51–53]. However, measurement of amine oxidase activity of IEX fractions – either containing or not containing ZAG – before adding recombinant AOC3 revealed no endogenous activity (Supplemental Fig. 2, C). Second, incubation of recombinant AOC3 with plasma of wt, AOC3 k.o. and ZAG k.o. mice enhanced AOC3 activity ~3-fold (Supplemental Fig. 1, D). Therefore, a plasma component found in all three genotypes must be responsible for enhancing AOC3 activity. Third, the IEX fractions lacking ZAG (D3 and D4) correspond to the major protein peak of the IEX profile, which mostly derives from albumin. Incubating recombinant AOC3 with fatty acid-free bovine serum albumin (BSA) also enhanced recombinant AOC3 activity to the same extent as murine plasma (Supplemental Fig. 2, A). Fourth, the literature describes a low molecular-weight plasma component (3.8 kDa), which in combination with lysophosphatidylcholine (LPC) boosts AOC3 activity [54]. LPC makes up to 4-20% of total plasma phospholipid content [55] and albumin is an important LPC storage protein [56]. Therefore, the IEX fractions lacking ZAG may contain the AOC3-activating plasma component, which is fully active in the presence of LPC. Notably, incubation of human lung-microsomal AOC3 with filtered and lyophilized human plasma (FLHP) enhances AOC3 activity up to 5-fold [54], which is similar to the effect of adding 200 μ l of the IEX fractions lacking ZAG to recombinant AOC3 (**Error! Reference source not found., B**).”

For not losing the focus on the main message of Fig. 4 panel D, I switched the diagram to a reduced form only comparing IEX fractions C12 and D1:

- The fig 5 also needs to be improved during revision: It demonstrates that ZAG addition clearly increases lipolysis slowly, but there no idea to quantify to what extend when compared to classical lipolytic agents , and as commented by the author , to what extend the positive control measured with β -adrenergic agonist on 1-2h might be useful for a lipolytic activity developed on 18 h?

Moreover in the different panels of Fig 5 , please indicate what is the tested concentration of the amines. Lastly, many typing errors and editorial pitfalls in the location of the data interpretation (either in fig legends, results or discussion) need to be corrected before publication.

For comparison of beta-adrenergic and ZAG induced lipolysis I integrated in the new MS another experiment which compared isoproterenol, ZAG, LJP1586 and GST (negative control) in parallel (Fig. 5, A). This clearly shows the short-time effect of isoproterenol compared to

ZAG. In Fig. 5, B of the new MS another time-dependence is integrated which also compares ZAG, GST and LJP1586 in parallel for twelve hours.

Both diagrams, Fig. 5 A and B, should illustrate that ZAG does not behave like a classical β -adrenergic agonist. This is described in the Results section of the new MS (**lines 240-250**):

“Most of the literature describes ZAG as an agonist of the β -adrenergic receptors, thereby stimulating downstream elements leading to an increase in lipolysis. To test this hypothesis, ZAG (50 $\mu\text{g/ml}$), GST (50 $\mu\text{g/ml}$), LJP1586 (10 μM) and isoproterenol (10 μM), a short-acting non-specific β -adrenergic agonist, were incubated with differentiated 3T3-L1 cells (**Error! Reference source not found.**, A). Compared to ZAG, isoproterenol significantly enhanced glycerol release already within the first thirteen minutes. ZAG, GST and LJP1586 showed no such effect. Incubating differentiated 3T3-L1 cells with ZAG (50 $\mu\text{g/ml}$), GST (50 $\mu\text{g/ml}$) and LJP1586 (10 μM) for several hours revealed that, although ZAG showed a lipolytic effect, this did not occur until twelve hours (**Error! Reference source not found.**, B). Therefore, ZAG definitely does not behave as a classical β -adrenergic agonist such as isoproterenol and another mechanism must be involved in ZAG-stimulated lipolysis, most likely involving AOC3.”

In the hope that my respond answers your questions and fulfills your ideas for improving the manuscript, I want to thank you once more for thoroughly revising my manuscript.

Yours sincerely,

Matthias Romauch, MD, PhD

Letter for Reviewer 2:

Dear Reviewer!

Thank you for reviewing the manuscript (MS). You are right to pay more attention on the results in the abstract, which substantially improves the MS. I will answer your questions by citing your comments first, followed by my answer.

- 1. Although the abstract is informative, the findings are not sufficiently summarised in it. Therefore, it should be re-written to summarise the results of the manuscript appropriately.**

According to your recommendation I re-wrote the abstract to summarize appropriately the results of the manuscript:

Abstract of the **old MS:**

“Zinc-alpha2-glycoprotein (ZAG) is a major plasma protein which is increased in chronic energy demanding diseases and serves as important clinical biomarker in the diagnosis and prognosis of the development of cachexia. Current knowledge suggests that ZAG mediates progressive weight loss through the β -adrenergic signaling in adipocytes resulting in activation of lipolysis and fat mobilization. This report defines a new role of ZAG acting as allosteric inhibitor of copper containing amine oxidase 3 (AOC3), a cell surface protein of adipocytes and endothelial cells. AOC3, also known as vascular adhesion protein 1 (VAP-1) and semicarbazide sensitive amine oxidase (SSAO), can affect both lipolysis and extravasation of immune cells. Here described observations suggest that ZAG acts on lipolysis in a β -adrenergic-indirect fashion and could modulate endothelial AOC3-dependent leukocyte transmigration.”

Is replaced with an extended abstract in the **new MS:**

“Zinc-alpha2-glycoprotein (ZAG) is a major plasma protein whose levels increase in chronic energy-demanding diseases and thus serves as an important clinical biomarker in the diagnosis and prognosis of the development of cachexia. Current knowledge suggests that ZAG mediates progressive weight loss through β -adrenergic signaling in adipocytes, resulting in the activation

of lipolysis and fat mobilization. Here, through crosslinking experiments, amine oxidase copper-containing 3 (AOC3) is identified as a novel ZAG binding partner. AOC3 – also known as vascular adhesion protein 1 (VAP-1) and semicarbazide sensitive amine oxidase (SSAO) – deaminates primary amines, thereby generating the corresponding aldehyde, H₂O₂ and HN₃. It is an ectoenzyme largely expressed by adipocytes and induced in endothelial cells during inflammation. Extravasation of immune cells depends on amine oxidase activity and AOC3-derived H₂O₂ has an insulinogenic effect. The observations described here suggest that ZAG acts as an allosteric inhibitor of AOC3 and interferes with the associated pro-inflammatory and anti-lipolytic functions. Thus, inhibition of the deamination of lipolytic hormone octopamine by AOC3 represents a novel mechanism by which ZAG might stimulate lipolysis. Furthermore, experiments involving overexpression of recombinant ZAG reveal that its glycosylation is co-regulated by oxygen availability and that the pattern of glycosylation affects its inhibitory potential. The newly identified protein interaction between AOC3 and ZAG highlights a previously unknown functional relationship, which may be relevant to inflammation, energy metabolism and the development of cachexia.”

2. There are several typos and grammatical mistakes, mainly in the methods section, in the manuscript that require attention.

The new MS is completely new edited with the help of “Cambridge academic manuscripts” (<http://www.cambridgeacademicmanuscripts.com/>). Figure legends and Figures are found at the end of the new MS and the supplementary material

In the hope that my respond fulfills your idea for improving the manuscript, I want to thank you once more for thoroughly revising my manuscript.

Yours sincerely,

Matthias Romauch, MD, PhD

Appendix B

Letter to Reviewer 2

Dear Reviewer,

I will answer your final concerns by citing you first followed by my answer:

The reviewer, although impressed by the quantity of work is still disappointed by the fact that the manuscript is signed only by one author, and especially respectfully disagrees with the style using 'I attempted to purify ...' or 'I tested whether ...' in lines 140, 179, 200 and elsewhere in the MS. Can at least these sentences be replaced by impersonal constructions?

- “To confirm the newly identified protein interaction, **I attempted** to purify AOC3 from *E. coli*.”

Is replaced with (lines 140-141):

“To confirm the newly identified protein interaction, **it was attempted** to purify AOC3 from *E. coli*.”

- “Subsequently, **I tested** whether recombinant mammalian ZAG inhibits endogenous AOC3 activity as effectively as that of recombinant AOC3.”

Is replaced with (lines 179-180):

“Subsequently, **it was tested** whether recombinant mammalian ZAG inhibits endogenous AOC3 activity as effectively as that of recombinant AOC3.”

- “Since recombinant ZAG inhibits endogenous AOC3, **I next asked** whether endogenous ZAG could inhibit recombinant AOC3.”

Is replaced with (lines 200-201):

“Since recombinant ZAG inhibits endogenous AOC3, **it was asked** whether endogenous ZAG could inhibit recombinant AOC3.”

- “Due to this inconsistency in how ZAG function is understood, **I followed** a more direct approach in this study to identify an interaction partner.”

Is replaced with (lines 395-396):

“Due to this inconsistency in how ZAG function is understood, a more direct approach **was followed** in this study to identify an interaction partner.”

- “Since plasma IEX fractions lacking ZAG enhanced recombinant AOC3 activity in a dose-dependent manner (Fig. 4, B), **I tested** whether plasma in general is able to enhance amine oxidase activity.”

Is replaced with (Supplemental Fig. 1, line 24-26):

“Since plasma IEX fractions lacking ZAG enhanced recombinant AOC3 activity in a dose-dependent manner (Fig. 4, B), **it was tested** whether plasma in general is able to enhance amine oxidase activity.”

Although deeply revised and improved, this version revealed small problems such as the lack of abbreviation for Sulfo SBED line 106:

Sulfo-SBED is now extended to (lines 106-107):

“Sulfo-SBED (Sulfo-N-hydroxysuccinimidyl-2-(6-[biotinamido]-2-(p-azido benzamido)-hexanoamido) ethyl-1,3'-dithiopropionate).”

and reviewer suspected a mismatch with the ref 46 in line 414 (not dealing with leukocyte adhesion as indicated).

Wrong references are replaced with the right reference #79:

Koskinen, K., Vainio, P. J., Smith, D. J., Pihlavisto, M., Ylä-Herttuala, S., Jalkanen, S. & Salmi, M. 2004 Granulocyte transmigration through the endothelium is regulated by the oxidase activity of vascular adhesion protein-1 (VAP-1). *Blood* **103**, 3388–3395. (doi:10.1182/blood-2003-09-3275).

Lastly, the long reference list does not exhibit homogeneous style, since still having full titles or abbreviations for journal names.

Therefore, small editorial issues should be treated before dissemination, probably at the step of proofreading.

All journals are now mentioned with their full names

I thank you very much for your support and for sorting out final mistakes.

Yours sincerely,

Matthias Romauch